# A Review of the Artificial Neural Network Models for Water Quality Prediction

**Yingyi Chen** [1,2,3,4,*] ⬤, **Lihua Song** [1,2,3] ⬤, **Yeqi Liu** [1,2,3], **Ling Yang** [1,2,3] **and Daoliang Li** [1,2,3,4]

1   Precision Agricultural Technology Integration Research Base (Fishery), Ministry of Agriculture and Rural Affairs, China Agricultural University, Beijing 100083, China; 874840197@cau.edu.cn (L.S.); liuyeqi@cau.edu.cn (Y.L.); zppayl@cau.edu.cn (L.Y.); dliangl@cau.edu.cn (D.L.)
2   College of Information and Electrical Engineering, China Agricultural University, Beijing 100083, China
3   National Innovation Center for Digital Fishery, China Agricultural University, Beijing 100083, China
4   Beijing Engineering and Technology Research Centre for Internet of Things in Agriculture, China Agricultural University, Beijing 100083, China
*   Correspondence: chenyingyi@cau.edu.cn; Tel.: +86-10-6273-8489

**Abstract:** Water quality prediction plays an important role in environmental monitoring, ecosystem sustainability, and aquaculture. Traditional prediction methods cannot capture the nonlinear and non-stationarity of water quality well. In recent years, the rapid development of artificial neural networks (ANNs) has made them a hotspot in water quality prediction. We have conducted extensive investigation and analysis on ANN-based water quality prediction from three aspects, namely feedforward, recurrent, and hybrid architectures. Based on 151 papers published from 2008 to 2019, 23 types of water quality variables were highlighted. The variables were primarily collected by the sensor, followed by specialist experimental equipment, such as a UV-visible photometer. Five different output strategies, namely Univariate-Input-Itself-Output, Univariate-Input-Other-Output, Multivariate-Input-Other(multi)-output, Multivariate-Input-Itself-Other-Output, and Multivariate-Input-Itself-Other (multi)-Output, are summarized. From results of the review, it can be concluded that the ANN models are capable of dealing with different modeling problems in rivers, lakes, reservoirs, wastewater treatment plants (WWTPs), groundwater, ponds, and streams. The results of many of the review articles are useful to researchers in prediction and similar fields. Several new architectures presented in the study, such as recurrent and hybrid structures, are able to improve the modeling quality of future development.

**Keywords:** ANNs; feedforward; recurrent; hybrid; water quality prediction

---

## 1. Introduction

Water quality plays an important role in any aquatic system, e.g., it can influence the growth of aquatic organisms and reflect the degree of water pollution [1]. Water quality prediction is one of the purposes of model development and use [2], which aims to achieve appropriate management over a period of time [3]. Water quality prediction is to forecast the variation trend of water quality at a certain time in the future [4]. Accurate water quality prediction plays a crucial role in environmental monitoring, ecosystem sustainability, and human health. Moreover, predicting future changes in water quality is a prerequisite for early control of intelligence aquaculture in the future [5]. Therefore, water quality prediction has great practical significance [6].

At present, there are many traditional water quality prediction methods, such as multiple linear regression (MLR) [7], auto-regressive integrated moving average (ARIMA) [8], etc. MLR is not able to detect a nonlinear relationship between water quality parameters because of its linear inherence [9].

The main drawback of ARIMA is the pre-assumption of the linear model [10]. During the model identification phase, the time series data must be checked to see whether they are stationary or not, because it is critical in creating the ARIMA model. In fact, traditional methods are not able to capture the non-linear [11] and non-stationarity [12] of water quality well due to their complex and sophisticated nature.

With the increase in data scale, traditional techniques cannot meet the demand of researchers. Owing to the improvement of computing power, artificial neural network (ANN) models, data-driven models, have been further developed. They can capture functional relationships among the water quality data from the examples [13]. When the underlying relationships of obtained data are difficult to describe, ANN models still work. Moreover, ANNs require fewer prior assumptions [14] and can achieve higher accuracy [15] compared with traditional approaches. In addition, ANNs are suitable for solving the non-linear and uncertain problems due to their similar characteristics with the brain nervous system [4], and have become a hotspot in water quality research [16].

ANNs are a family of models inspired by biological neural networks [17] which specifically refers to the human brain [18], a kind of central nervous system of animals. In general, ANN can be represented as a system of interconnected "neurons" [19] which form the basis of neural network operation. Weight parameters and activation functions are part of the neurons [20]. ANNs are generally divided into three layers of input, hidden and output. When neurons receive information from different inputs, they obtain nonlinearity through activation functions. ANN models depend heavily on the quantity of data [21]. Therefore, it is not recommended to use relatively small data sizes for predictors (inputs). This is because some useful information is lost in short-term data, which may lead to poor prediction results [3]. In addition, data dividing is a necessary step in the modeling process. Furthermore, choosing the training algorithm to calibrate the model parameters (e.g., connection weights) is a vital step so that the network can approximate complicated non-linear input-output relationship [10]. The Levenberg–Marquardt [22] algorithm and the back-propagation (BP) algorithm [23] are the most commonly used algorithms.

ANN models architectures determine the number of connection weights and the way information flows through the network [20]. The most widely used architecture is Multilayer Perceptron (MLPs) with only three layers in many types of feedforward ANNs. Radial Basis Function neural networks (RBFNNs) [24], General regression neural networks (GRNNs) [25] and Extreme learning machines (ELMs) [5] are three typical feedforward ANNs. A Long Short-Term Memory (LSTM) neural network is an improvement of recurrent neural networks (RNNs), which aims to address the well-known vanishing gradient problem [26]. The hybrid models in this review are three classes: model-intensive, technique-intensive, and data-intensive [27]. The emerging frameworks, such as Convolutional Neural Network (CNN) [28], widely used in the field of the image, are also included in this review.

In this review, ANN models for water quality variables prediction are summarized. Previous reviews [20,27,29] about ANNs are more concerned about the water quantity (e.g., flow and rainfall-runoff) prediction, while less attention has been paid to water quality prediction (e.g., Suspended solids (SS)), and the major scenarios they investigated are river systems. At the same time, previous reviews care about the development of the model while ignoring the output strategies between input(s) and output(s) in a given prediction task. To overcome the limitations above, this review focuses on the use of ANNs methods for water quality prediction, with more water quality variables investigated than previous reviews, which are mainly divided into three categories, namely chemical, biological and physical variables [30].

The research scenarios include not only the river system that was the focus of the previous review, but also reservoirs, lakes, wastewater treatment plant (WWTP), groundwater, etc. It must be pointed out that the review did not consider drinking water systems. The reason for this is that drinking water is a system that includes source, treatment, and distribution, and should be considered as an independent branch or subject for systematic research [30]. In addition to the increased number of water quality variables reviewed and broader research scenarios, this review also summarizes five

output strategies. The period of the investigated papers covered was from 2008 to 2019. This period was chosen as it follows on from the period covered in the review by [27] (i.e., 1999–2007). The review is organized as follows. Section 2 presents the process of the paper collection. Section 3 describes three basic model structures in water quality prediction. In Section 4, the applications of artificial neural networks in water quality are surveyed. Then, Section 5 represents the results of this review. Finally, the discussions are given in Section 6. All the abbreviations are mentioned in Table 1.

**Table 1.** The abbreviations in this review.

| Abbreviations | Full Name | Abbreviations | Full Name | Abbreviations | Full Name | Abbreviations | Full Name |
|---|---|---|---|---|---|---|---|
| AH | air humidity | EC | Electrical conductivity | ORP | Oxidation reduction potential | TCC | total chromium concentration |
| AODD | August, October, December, data | Evap | evaporation | Q | discharge | TIC | total iron concentration |
| AP | air pressure | FTT | flow travel time | pH | Pondus Hydrogenii | TAC | total anions and cations |
| AT | air temperature | Fe | iron | Precip | precipitation | TNs | total nutrients |
| As | Arsenic | F | flow | P | phosphate | TA | total alkalinity |
| B | boron | HCO$_3$ | bicarbonate | RH | relative humidity | TP | total phosphorus |
| BOD | Biochemical Oxygen Demand; | HA | Hydrogenated Amine | RP | Redox potential | Tur | turbidity |
| C | carbon | ICs | ionic concentrations | RO | runoff | TDS | total dissolved solids |
| Cl | chloride | K | potassium | RF | rainfall | TN | total nitrogen |
| Cu | Copper | Lon | longitude | RainP | Rainy period | TH | total hardness |
| Ca | calcium | Lat | latitude | SR | solar radiation | TOC | total organic carbon |
| CO$_3{}^{2-}$ | Carbonate | LV | lake volume | Sth | sunshine time hours | TSS | total suspended solids |
| Coli | Coliform | MDHM | month, day, hour, minute | SD | transparence | VP | volatile phenol |
| COD | Chemical Oxygen Demand | Mn | manganese; | SAR | sodium absorption ratio | WL | Water Level |
| COD $_{Mn}$ | permanganate index | Mg | magnesium | SM | Soil Moisture | WT | water temperature |
| Chl-a | Chlorophyll a | Na | sodium | ST | soil temperature | WS | wind speed |
| DO | dissolved oxygen | Ns | nutrients | SO$_4$ | sulphate | WD | wind direction |
| DOY | day of year | NO$_2$ | nitrite | S | salinity | YMDH | the year numbers |

## 2. Methods

This review focuses on the application of ANNs to water quality variables prediction excluding drinking water from 2008 to 2019. The papers to be reviewed were selected using the following steps:

1. First, we identified ANN-related papers in influential water-related and environmental-related journals to ensure that high-quality papers are included in the review. These papers are mainly from journals whose subjects are environmental science and ecology, water resources, engineering and application.

2. Thereafter, a keyword search of the ISI Web of Science was then conducted for the period 2008–2019 using the keywords; water quality, river, lake, reservoir, WWTP, groundwater, pond, prediction, and forecasting, accompanied by the names of ANN methods (one or more), such as neural network, MLP, RBFNN, GRNN, RNN, to name but a few.

3. Then, through the search process from 1 to 2, 151 articles in English relevant to our focus were selected. The basic information of the papers, including authors (year), locations, water quality variables, meteorological factors, other factors, output strategy, data size, time step, data dividing, methods, and prediction lengths are provided in Appendix A.

## 3. Three Basic Model Structures in Water Quality Prediction

In this review, the model architecture refers to the overall structure and manner of how information flows from one layer to another. The three model architectures include feedforward, recurrent networks, and hybrid models (see Figure 1) [31]. In addition to categorizing each architecture, Table 2 summarizes the foundation and advantage(s) of the development model structure.

### 3.1. Feedforward Architectures

The term 'feed-forward' means that a neuron connection only exists from a neuron in the input layer to other neurons in the hidden layer or from a neuron in the hidden layer to neurons in the output layer. However, the neurons within a layer are not interconnected [9]. MLPs with only three layers are the most widely used architectures [59] in many types of feedforward ANNs (see Figure 2), followed by BPNNs [37] which use the back-propagation algorithms to train networks. Other commonly used feed-forward network architectures in water quality prediction include TDNNs [36], RBFNNs [60], GRNNs [61], WNNs [62], ELMs [5], CCNN [63] and MNN [50].

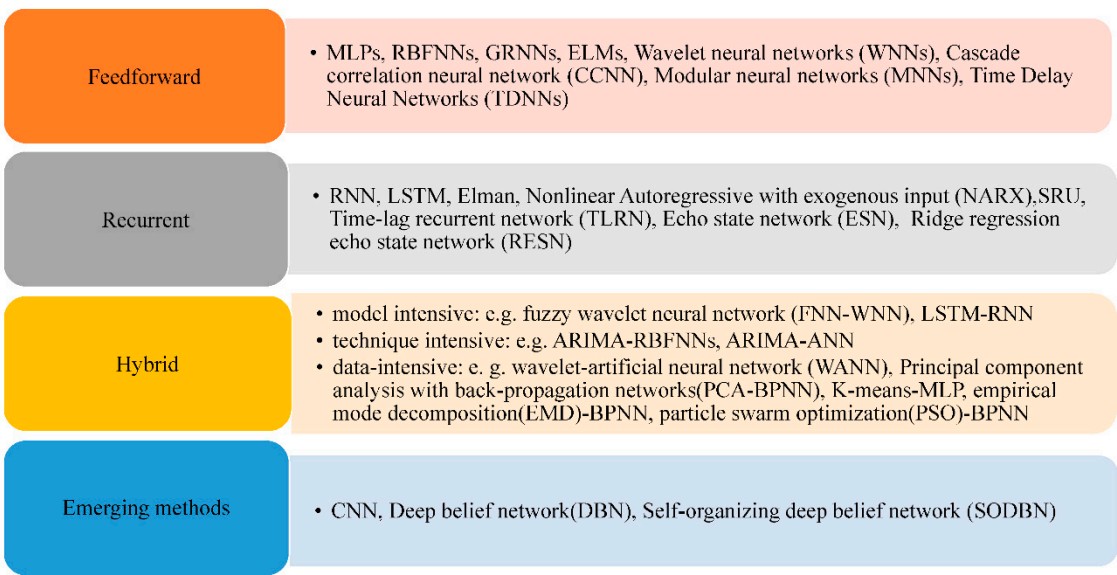

**Figure 1.** Three main model architectures in the reviewed papers.

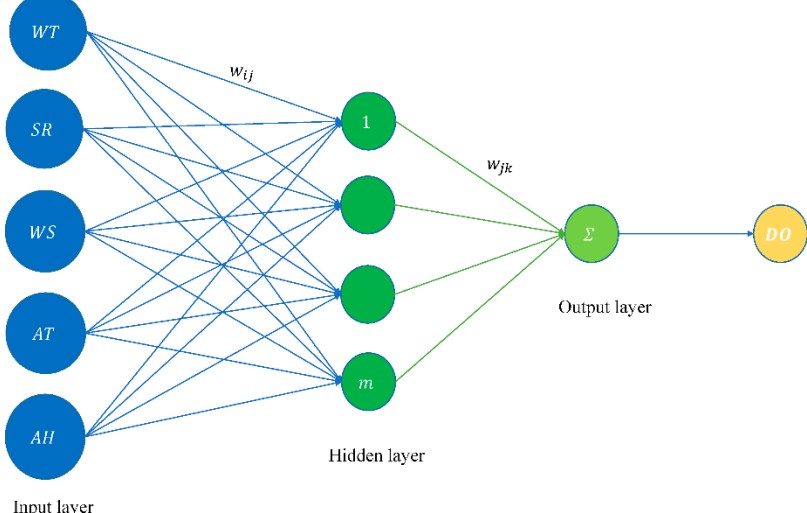

**Figure 2.** The common architectures of MLPs.

**Table 2.** The developments and advantages of different ANNs architectures.

| Categories | Structure(s) | Advantage(s) | Reference(s) |
|---|---|---|---|
| MLPs | They are based on an understanding of the biological nervous system | Solving the nonlinear problems | [19,23,30,32–35] |
| TDNNs | They are based on the structure of MLPs | Using time delay cells to deal with the dynamic nature of sample data | [36] |
| RBFNNs | The structure of RBFNNs is similar to the MLPs The radial basis activation function is in the hidden layer | To overcome the local minimum problems | [5,18,37,38] |
| GRNNs | A modified form of the RBFNNs model There is a pattern and a summation layer between the input and output layers | Solving the small sample problems | [24,39–43] |
| WNNs | Wavelet function replace the linear sigmoid activation functions of MLPs | Solving the non-stationary problems | [16,44] |
| ELMs | The structure of ELMs is similar to the MLPs Only need to learn the output weight | Reducing the computation problems because the weights of the input and hidden layer need not be adjusted | [31,45–48] |
| CCNN | Start with input and output layer without a hidden layer | A constructive neural network that aims to solve the problems of the determination of potential neurons which are not relevant to the output layer | [49] |
| MNNs | A special feedforward network Choosing the neural network which have the maximum similarity between the inputs and centroids of the cluster | Solving the problem of low prediction accuracy | [30,50] |
| RNNs | The RNNs are developed with the development of deep learning | Solving the problems of long-term dependence which are not captured by the feedforward network | [12,31,38,51,52] |
| LSTMs | Its structure is similar to RNNs Memory cell state is added to hidden layer | Addressing the well-known vanishing gradient problem of RNNs | [15,26,45,53,54] |
| TLRN | Its structure is similar to MLPs It has the local recurrent connections in the hidden layer | Reducing the influence of the noise and owning the advantage of adaptive memory depth | [55] |
| NARX | Sub-classes of RNNs Their recurrent connections are from the output | Solving the problems of long-term dependence | [12] |
| Elman | A context layer that can store the internal states is added besides the traditional three layers | It is useful in dynamic system modeling because of the context layer | [3] |
| ESN | Different from the above recurrent neural networks The three layers are input, reservoir, and readout layer | To overcome the problems of the local minima and gradient vanishing | [3] |
| RESN | They are based on the structure of ESN which has a large and sparsely connected reservoir | To overcome the ill-posed problem existing in the ESN | [3] |
| Hybrid methods | The combination of conventional or preprocess methods with ANNs The internal integration of ANN methods or | Exploring the advantages of each methods | [56] |
| CNN | Input, convolution, fully connection, and output layers | An emerging method to solve the dissolved oxygen prediction problem | [57] |
| SODBN | They are based on the structure of DBN whose visible and hidden layers are stacked sequentially | Investigating the problem of dynamically determining the structure of DBN | [58] |

TDNNs is a subclass of MLPs that learns temporal behavior from continuous past and present signals [36]. The major difference between RBFNNs and MLPs is that the hidden layer of RBFNNs is self-organizing while the latter is not, although the structure of RBFNNs is similar to MLPs. As the center of RBF, the training weights can be defined by a clustering algorithm. For example, the k-means algorithm is a commonly used one [24]. GRNNs is a modified form of the RBFNNs model, but it differs from RBFNNs in structure. Patten and summation layers are located between the input and output layers [27]. The training between the input and pattern layer of GRNNs is equivalent to the research on the input and hidden layer of the RBFNNs. WNNs have made some changes based on the traditional MLPs, in which the non-linear sigmoid activation functions is replaced by the Morlet wavelet function commonly used in the WNNs hidden layer. Therefore, WNNs are suitable for solving non-stationary time series problems [64]. The biggest innovation of ELMs is the random selection of hidden nodes and the use of a least squares method to determine the output layer weight. CCNN is different from the above feedforward networks because it constructs the neural network without a hidden layer at first and automatically adds hidden units instead of fixing the network architectures and then training the weights and thresholds. The first step of MNN, a special feedforward network, is data clustering using the fuzzy c-means method [65]. The second step is updating the clusters by adding the new datasets. To achieve better prediction accuracy, a neural network with the maximum similarity between the inputs and centroids of the cluster is chosen.

### 3.2. Recurrent Architectures

Compared with feedforward ANNs, RNNs differs in that neurons within a layer are interconnected and allow feedback [53]. Different types of RNNs are developed so that the neural networks have better memory ability (see Figure 1). LSTM, an improvement over RNN, adds a processor called "memory cell state" to its hidden layer to determine whether the information is useful or not [66], and this is also suitable for SRU (Simple Recurrent Unit) [67]. Furthermore, the forget gate also determines what information should be discarded from the cell state [66]. TLRN has a similar structure to MLPs, but has local recurrent connections in the hidden layer (see Figure 3), with the advantages of low noise sensitivity and adaptive storage depth [55]. NARX networks are also sub-classes of RNNs and can be utilized to establish a long-term temporal relationship. The recurrent connections of NARX networks come from the output (see Figure 3) [12]. In addition to the input, hidden, and output layers, the Elman neural network has a context layer to store the internal states [3]. The Elman neural network is sensitive to the historical information of inputs because of the self-connections of the context nodes (see Figure 3). The three layers of ESN are different from the above recurrent neural networks. The three layers are input, reservoir, and readout layer. The feature of the reservoir layer is randomly and sparsely connected. The echo state property whose internal states are particularly dependent on the inputs is the key to the ESN. To overcome the ill-posed problem existing in the ESN, an RESN method using the ridge regression algorithm instead of linear regression to calculate output weights is proposed [38].

### 3.3. Hybrid Architectures

There is a growing tendency to use hybrid ANNs models, which play a huge role in modeling, for their ability to integrate with other conventional and more advanced modeling techniques [68], to create flexible and efficient models in recent years (see Figure 1). Hybrid models are divided into three categories, namely model-intensive, technique-intensive, and data-intensive [27]. The model-intensive approaches model the sub-components of the whole physical system and aggregate the overall response of each model. Relevant forms, such as LSTM-RNN [26] or FNN-WNN [69], are model-intensive methods. The core of the technique intensive methods is to develop a modeling framework that is able to take advantage of different technologies. Methods that combine ensemble approaches [32] or time series models that remove trends or periodicities like Autoregressive Integrated Moving Average-Radial Basis Function neural networks (ARIMA-RBFNNs) [70] or ARIMA-ANN [71] are

technique-intensive methods. In this review, data-intensive approaches are to combine different technologies to preprocess the data. Wavelet analysis approaches such as WANN [72] can provide some useful information about the physical structure of the data. ANNs models the approximation and details component from the discrete wavelet transformation (see Figure 4). Dimensionality reduction methods such as PCA can reduce the dimension of the input data space to prevent redundancy [73]. Then, ANNs models some aggregative indices obtained by PCA (see Figure 4). Clustering methods [50] such as K-means-MLP [43] identify the data belonging to a particular class. Other data-intensive approaches include decomposition [5] and evolution-related [16] methods. ANN models the Intrinsic Model Function (IMF) obtained from the decomposition of complicated signals.

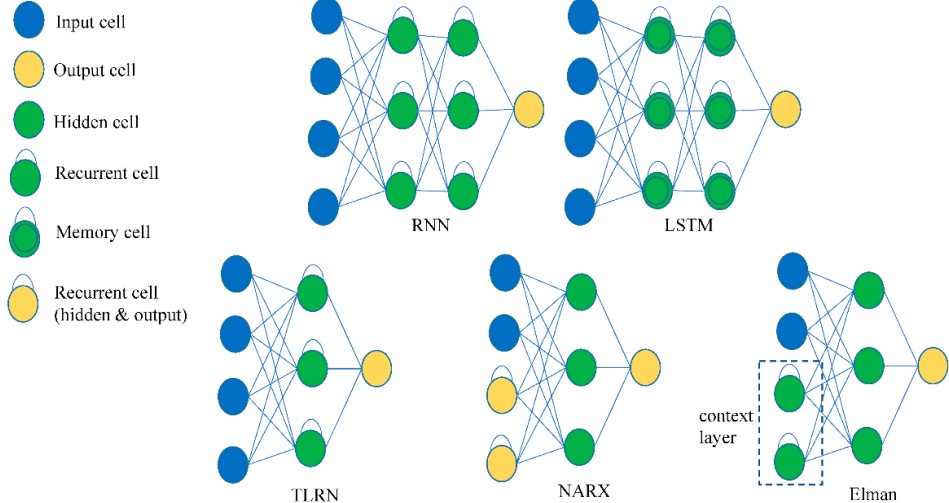

**Figure 3.** Five categories of recurrent model architectures.

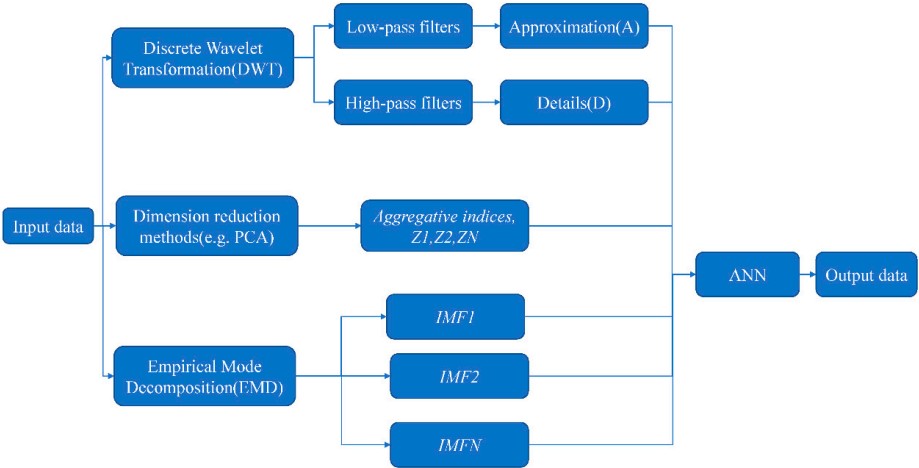

**Figure 4.** The modeling process of three data-intensive approaches.

### 3.4. Emerging Methods

CNN is a feed-forward neural network, primarily used in the image field. Input, convolution, pooling, full connection, and output layers are the basic elements of the traditional CNN. In recent years, CNN has been used as an emerging method in water quality prediction. The operation of convolution can be implemented more than one time to reveal the relationship between the parameters hidden in the input matrix [57]. However, since the purpose of the prediction model is to extract potential factors rather than simply raise the convolutional layer's results to a higher level, the pooling layer is removed (see Figure 5). In the meantime, the number of calculations can be reduced.

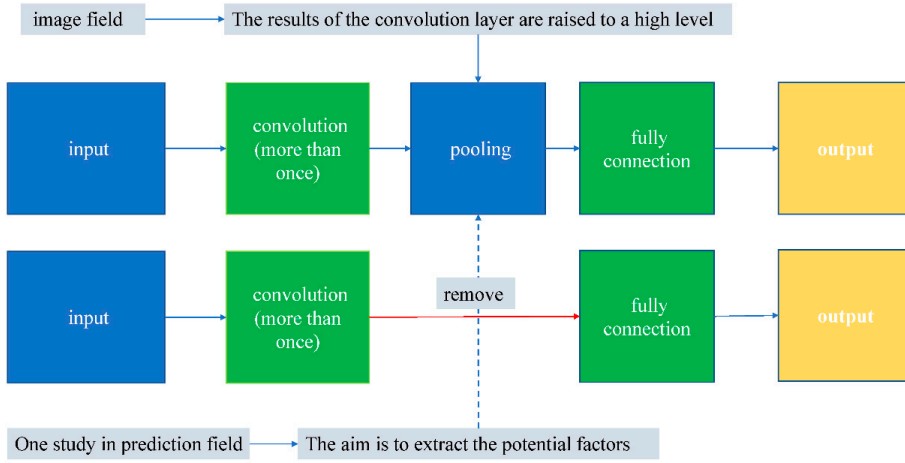

**Figure 5.** The architecture of a Convolutional Neural Network.

Deep belief network (DBN) is a kind of neural network based on deep learning which is similar to feedforward structure and has been widely used in recent years. The blue virtual box in Figure 6 shows several visible and hidden layers, stacked in order to make up the DBN [74]. However, the researches about dynamically determining the structure are seldom investigated. To overcome the limitations above, a SODBN has been proposed. The structure of the SODBN is not determined by artificial experience but the automatic growing and pruning algorithm (AGP) [58]. Especially, the hidden layers and neurons are changed by the AGP at first. Then, the weights of the SODBN are continuously adjusted in the process of self-organization. Finally, some aspects of network performance, such as running time and prediction accuracy have been improved.

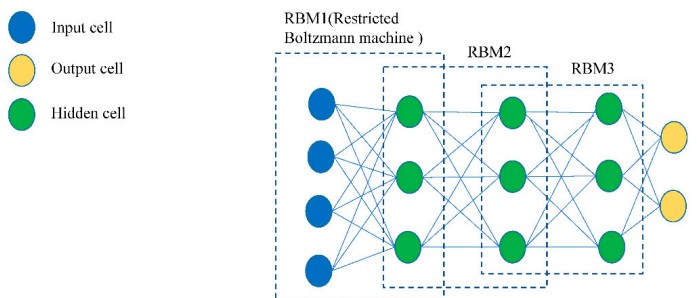

**Figure 6.** The architecture of deep belief network.

## 4. Artificial Neural Networks Models for Water Quality Prediction

From 2008 to 2019, the use of the ANN technique has been very popular in the field of water quality prediction. Many researchers have utilized ANNs to model and predict water quality. Dogan et al. [75] adopt ANN to predict the BOD, which is difficult to measure and needs at least five days to get the final results in WWTP. Results showed that COD was the most effective variables on BOD estimation after conducting the sensitivity analysis. Elhatip and Kömür [76] revealed that ANN techniques depend on using more input data to solve the water quality problems, although they did not illustrate the size of the appropriate datasets. Palani et al. [40] tested MLP and GRNN models with various input selected by stepwise constructive methods for multistep prediction of S, DO, and Chl-a. They pointed out that the limited data set was one of the drawbacks of their research and encouraged others to collect more data to recalibrate and revalidate the model. Wang et al. [19] employed a typical three-layer of MLP structure [77–89] with the BP algorithm to achieve Chl-a prediction. They divided the dataset into training (75%) and testing parts (25%). Results indicated that ANNs could establish a stable and effective model for Chl-a prediction. This result is also suitable for other parameters

prediction. Yeon et al. [90] evaluated ANN, MNN, and adaptive neuro-fuzzy inference system (ANFIS) performance in 1-h and 2-h ahead prediction of DO and TOC. They added Q to inputs because rainfall affected the water quality prediction. It was found that using the Levenberg–Marquart algorithm to train the MNN could provide the least error and better results. Dogan et al. [91] divided the data into training (60%), validation (20%), and testing sets (20%). They adopted a sensitivity analysis method to find out the important water quality parameters and excluded fewer influence variables, resulting in a compact network. Miao et al. [92] used BPNN to COD and ammonia nitrogen ($NH_3$-N) prediction. The whole datasets were normalized at first and then divided into training (80%) and testing (20%) sets. The sigmoid transfer function that can establish the random nonlinear map between inputs and outputs were adopted. Oliveira Souza da Costa et al. [93] divided the data into training (50%), validation (25%), and testing sets (25%). Shen et al. [94] employed a golden section method to select the hidden layer nodes of BPNN. Singh et al. [95] investigated the partition approach in evaluating the relative importance of eleven environmental variables to the output layer. They divided the datasets into training (60%), validation (20%), and testing sets (20%). Results showed that the predicted values of the ANN model were close to the measured value. Yeon et al. [96] combined Precip and Q to realize a one-step prediction of Q. Then, the connected system utilized the prediction value of Q and historical TOC to fulfill the one-step prediction of TOC. Finally, the connected system had better performance than a single ANN model. Zuo and Yu [97] pointed out that ANN models could process complex and multivariable problems. Akkoyunlu and Akiner [98] verified the feasibility of ANN technique, data-driven models, in predicting DO. Results showed that the ANN method was superior to the nonlinear regression (NLR) technique. Chen et al. [99] scaled the datasets to lie between 0 and 1 [9,16,59,62,100–104] so that it could be compatible with the sigmoid transfer functions used in the hidden layer and applied the constructive and pruning of stepwise methods that aim to maximize the model's performance through a constant adjustment to surface water quality prediction. Markus et al. [105] purely relied on a trial-and-error approach to determine the model structure and dividing the data into training (50%) and testing sets (50%). Result found that ANN could improve the forecast accuracy of $NO_3$ compared with previous studies. Merdun and Çinar [106] preprocessed the data set by normalization and moving average techniques. They improved the representation of the acquisition data through a data preprocessing technique. Ranković et al. [107] used a sensitivity analysis method to determine the influence of input variables on outputs and found out that 15 hidden neurons gave the best choice. Zhu et al. [108] not only predicted the water quality using ANN models but also introduced a remote wireless monitoring system. Banerjee et [109] checked that ANN models were an accurate alternative to the numerical methods. They used quick propagation algorithm to realize super linear convergence speed. Han et al. [110] demonstrated the effectiveness of a flexible structure RBFNN which using neuron activity and mutual information (MI) to add or remove hidden neurons to reduce network complexity and improve computational efficiency. The connected weights are trained by an online learning algorithm. Zare et al. [10] used a UV-visible photometer to measure the $NO_3$ concentration in the laboratory.

Asadollahfardi et al. [111] utilized Q to forecast TDS when TDS was not available. Al-Mahallawi [77] revealed that the reason why ANN models could model complex water quality phenomena was that they provided a non-linear function mapping from input to corresponding network outputs. Ay and Kisi [112] divided the data into training (50%), validation (25%), and testing sets (25%). In the three parts of data division, the validation set can be implemented more than once to monitor whether the model is overfitting or not. Comparison results showed that the RBNN model performed better than MLP in DO prediction. Baek et al. [50] chose the neural network of MNN, which has the maximum similarity between the inputs and centroids of the cluster, to solve the problem of low prediction accuracy. They introduced Gradient descent with momentum and Levenberg–Marquardt backpropagation (TRAINLM) to train the neural network. Bayram et al. [79] used the one-year Tur data whose time step is fortnightly to achieve the prediction of SS. Gazzaz et al. [113] scaled the data into the scope between 0 and 1 and utilized cross-validation to improve the generalization ability

and limit the overfitting problem. Cross-validation was suitable for the situation where the size of the training data was small or the number of parameters in the model was large. Overfitting refers to the situation that when the error on the training set is driven to a very small value, the test data are presented to the network with a large error. That means the network has memorized the training examples, but it has not learned to generalize to new situations. Hong [78] took the AT, AP, WD, and WS variables measured by meteorological station into account. They divided the data samples into training (70%) and testing (30%) sets. Results indicated that MLP also could deal with large data samples. Liu and Chen [114] recorded the location information to complete the three-dimensional DO prediction. Tota-Maharaj and Scholz [22] assessed the influence of bp, Levenberg–Marquardt, Quasi-Newton, and Bayesian Regularization algorithms on BOD prediction. Results showed that the combination of bp and ANN had low minimum statistical errors. Kakaei Lafdani et al. [115] firstly used M-test to obtain several data points through the winGamma software. Then, the genetic algorithm (GA) method was implemented to make the best combination which extracted from a list of possible inputs as inputs. Karakaya et al. [116] conducted research, namely temporal partitioning, to divide the data into diel, diurnal, and nocturnal in order to obtain continuous records, and chose MLP as a prediction model. Antanasijević et al. [117] utilized Monte Carlo simulation (MCS), a sensitivity analysis method that involves repeatedly generating a probability distribution of random input values, to ultimately create an ANN model with fewer inputs. Moreover, other input selection techniques include correlation analysis and genetic algorithm were tested. Chen and Liu [118] utilized sigmoid and linear transfer function in the hidden and output layer, respectively. Results showed that ANFIS and BPNN could predict DO with reasonable accuracy. Han et al. [119] adopted linear interpolation whose data increment was calculated by the slope of the assumed line to fill the missing data. Then, hierarchical ELM based on a hierarchical structure was chosen to model the DO, pH, and SS. The advantage of hierarchical ELM is able to learn sequential information online. Results demonstrated the effectiveness of the proposed methods. Researchers tended to divide the training set data into 70% to 90% of the total data [39,42,49,52,72,120–127]. Iglesias et al. [35] divided the data into training (90%) and testing sets (10%). Then, they applied three typical MLP architectures to complete the Tur prediction whose inputs were NH$_3$-N, EC, DO, pH, and WT. Klçaslan et al. [128] randomly divided the datasets and pointed out that when the data tended to be roughly periodic after a year, the time length of data acquisition, covering a long period such as a year or more was highly recommended in order to capture long-term variation. Yang et al. [129] found the most significant parameters by using analysis of variance (ANOVA) techniques. Result indicated that rainfall records were the most significant parameters for turbidity forecasting. Khashei-Siuki and Sarbazi [130] took the normalization step to control the scale of each feature, in the same range in case the difference of the order of magnitude will lead to the dominance of larger attributes thereby slowing down the iterative convergence. However, they did not give clear details about normalization. Gholamreza et al. [36] used time delay cells of TDNNs, designed based on the structure of MLPs, to deal with the dynamic nature of sample data. Then, they applied factor analysis to select the model inputs. Results illustrated that TDNN with 2 hidden layers of 15 neurons in each of the layers was the best architecture. Nourani et al. [9] provided a new solution to EC and TDS prediction. When the predictive variables were not available, researchers could realize the final predictions through modeling other relevant variables. They utilized monthly meteorological data RF, RO, and WL to forecast EC and TDS due to the lack of historical records of outputs. Zounemat-Kermani [82] introduced a Quasi-Newton method, Broyden–Fletcher–Goldfarb–Shanno (BFGS), to train the parameters of MLP in SS forecasting. Hameed et al. [60] conducted the sensitivity analysis of the obtained data and scaled it to between 0.1 to 0.9. Results indicated that RBFNN could achieve high-performance accuracy. Heddam and Kisi [47] utilized open-source data from Eight United States Geological Survey stations (USA) and preprocessed the data by standardization method. Several ELM models are applied for DO prediction. Yousefi [131] discussed the Garson method to find the relative importance of each input variable. Results indicated that including meteorological and hydrologic variables could improve the accuracy of the models

with fewer influential variables. Elkiran et al. [32] and Najah et al. [132] demonstrated the feasibility of the ANFIS method in predicting river water quality. This model overcame the shortcomings of ANN models such as overfitting and local minima, and combined fuzzy logic with ANN to provide a method to solve uncertain problems. Sinshaw et al. [133] took interrelated and easily measurable parameters of pH, EC, and Tur, as inputs to realize TN and TP predictions.

Liu et al. [3] pointed out that if more historical data were available [15], ANN models may provide better predictions than a relatively small data set. Antanasijević et al. [41] tested the performance of RNN, GRNN, and MLP in small samples prediction. Results indicated that the error of RNN in test data was less than 10%. Besides, the error of GRNN was lower than MLP. Evrendilek and Karakaya [55] deleted the missing data directly. Then, discrete wavelet transforms (DWT) with the orthogonal wavelet families was applied to denoise the data measured by proximal sensors. The result indicated that the modeling effect of using TLRN to the data after noise reduction was superior to TLRN, TDNN, and RNN. Chang et al. [12] attempted to use NARX, a dynamic neural network, to model ten-year seasonal water quality data. Then, 42-fold cross-validation was used to divide the data. Results demonstrated that the NARX network outperformed BPNN because it could capture the important dynamic features of TP data. Wang et al. [6] tested the prediction performance of LSTM, BPNN, Online sequential (OS)-ELM in DO, and TP. The results indicated that LSTM was more accurate and generalizable than the above feedforward ANNs. Zhao et al. [38] used an improvement of the ESN, namely RESN, to predict the BOD and TP. This new method used the ridge regression algorithm to calculate the output weights to solve the ill-posed problem existing in the ESN. Hu et al. [66] fully preprocessed the acquired water quality data. They firstly imputed, corrected, and denoised the data by using linear interpolation, smoothing which could attenuate high-frequency signals, and moving average filtering techniques. Then, correlation analysis, which belongs to analytical methods, was carried out. The LSTM was adopted for model establishment. Experimental results showed that the prediction accuracy was high and could reach 98.97% and LSTM was suited for long-term prediction. J. Liu [67] introduced Back-propagation through time (BPTT) to train the SRU model. The main difference between SRU and RNN is the "cell state" part added in the hidden layer. They proposed an Improved mean value method to solve the breakpoint phenomenon of the mean value method and the linear interpolation method. Results showed that the prediction error was small, within the range of 1%. Lim et al. [53] converted the irregular data into daily data by using a linear interpolation method and provided a solution to abnormal data identification. They used a fixed threshold method to set the upper and lower threshold ranges and proved that linear interpolation had better robustness than spline interpolation, nearest-neighbor interpolation, and cubic interpolation according to model results when water quality changed dramatically. Results showed that the removal of abnormal data beyond the threshold value could preliminarily improve the data convergence.

Partal and Cigizoglu [134] decomposed the measured SS data into wavelet components via DWT. The DWT-ANN method could more accurately approximate the peak values, which have lesser distributions compared with non-peak values. Anctil et al. [135] applied MLP to forecast daily SS and NO3 without considering missing data. They applied a self-organizing map (SOM), a stratified method, to construct a topological map to visualize the clustered input variables, thereby ensuring that the statistical properties of the subsets were similar. Levenberg–Marquardt algorithm [24,136–139] and Bayesian methods were conducted to train the network. Results showed that ANN models could achieve high accuracy. Sahoo et al. [140] used the SR and AT meteorological data to achieve the WT prediction. They introduced micro-genetic algorithms (u GA), a creep mutation in small populations, to update the weights. Wu et al. [141] reported that the GA-BP algorithm whose relative errors were below 35% was more suitable for TP, TN, and Chl-a prediction than simple multivariate regression analysis. Kişi [142] utilized neural differential evolution (NDE) models, a combination of neural networks and differential evolution approaches, to model SS. The result showed that NDE has a low mean square error. Ömer Faruk [71] investigated the performance of ARIMA-ANN in WT, DO, and B prediction. Afshar and Kazemi [143] combined PSO and ANN methods in water quality parameter

prediction. Han et al. [1] used cross-correlation and mutual information to select the input to achieve the prediction of BOD and DO, respectively. The conjugate gradient algorithm was carried out to train the model. Areerachakul et al. [144] presented two cluster technique, namely K-means, fuzzy c-means (FCM) in DO prediction. Results indicated that the performance of hybrid methods was better than single models. Y. Wang [64] designed a missing–refilling scheme which divided the data into incidental missing (ID) and structural missing (SD). Then, a temporal exponentially moving average was applied to fill the missing data. They investigated the time relationship of the DO, $NH_3$-N univariate time series using a bootstrapped wavelet neural network (BWNN). Aleksandra and Antanasijevi [42] used the databases of the European Statistical Office and World Bank to complete the BOD prediction. Ay and Kisi [43] integrated k-means clustering and MLP in daily COD concentration modeling by using SS, pH, and WT. Result indicated that this hybrid methods performed better than MLP, RBFNN, and two different ANFIS approaches (subtractive clustering and grid partition). Ding et al. [120] collected 23 water quality parameters and considered the problems of data dimensionality. Therefore, the PCA techniques was used to compress the original data into 15 aggregative indices. Then, the GA approach was applied to optimize the parameters of BPNN. The result showed that the average prediction accuracy was up to at least 88%. Gazzaz et al. [145] developed a data mining method, namely re-sampling, to solve the unbalance problem. Heddam [146] recommended collecting more than one-year water quality data, because they wanted to include all four seasons in the validation and testing phases. Liu et al. [147] proposed a hybrid model, namely empirical mode decomposition (EMD)-BPNN. BPNN predicted each sub-series which are IMFs and the residue decomposed by EMD. The results demonstrated that a hybrid model could capture the non-stationary characteristics of WT after EMD. Qiao et al. [44] scaled the datasets between -1 and 1 and then used phase space reconstruction (PSR) of chaos theory to extract much more information from BOD datasets. Results showed that the hybrid model, namely chaos theory-PCA-ANN, had high prediction accuracy. Sakizadeh et al. [73] applied early stopping which is fit for small networks and datasets to determine the model structure.

Yu et al. [148] utilized 5-fold cross-validation to divide the data and applied RBFNN to fuse data from multiple sensors. The convergence rate and the solution accuracy could be improved through the variant of PSO (IPSO). The comparison of prediction results validated the effectiveness of the hybrid model. Zhao et al. [149] converted the signal into an output linear system by the Kalman filter. The result showed that this hybrid method was a good and effective approach to water quality prediction. Huang et al. [69] simulated the nonlinearity of data by the combination of the neural network, fuzzy logic, wavelet transform, and the GA. Results showed that this hybrid model could handle the problems of data fluctuation. Li et al. [123] adopted the most extreme form of K-fold cross-validation, namely leave-one-out cross-validation to divide the datasets. Zhang et al., 2017 [16] divided the dataset into training (98%) and testing sets (2%) and adopted the PSO algorithm to accelerate the training speed of WNN. Karaboga proved that artificial bee colony (ABC) algorithms were more precise than GA and PSO [150]. Chen et al. [4] proposed an improved method of ABC (IABC) which added the optimal and global optimal solution to the updated formulas. The result indicated that the limitation of the method above was that water quality data needed to obey the normal distribution appropriately. Li et al. [54] used sparse auto-encoder (SAE) to pre-train the hidden layer data because SAE contained deep latent features. Qiao et al. [58] determined the structure of DBN by growing and pruning algorithms instead of artificial experience (SODBN). Results showed that SODBN could short running time and improve accuracy. Ta and Wei [57] applied Adam optimization method which could handle sparse gradients on noisy problems to train the parameters of CNN. Zhou et al. [151] focused on the Improved Grey Relational Analysis (IGRA) method which calculated the similarity and proximity by relative area change ratio. Fijani et al. [5] used variational mode decomposition (VMD) algorithm to decompose the highest frequency component produced by a complete ensemble empirical mode decomposition algorithm with adaptive noise (CEEMDAN). ELM was applied for modeling. Results indicated that this hybrid model could reduce error whether in root mean square or mean absolute error. Jin et al. [152] proposed an improvement variant namely

improved genetic algorithm (IGA) to avoid the situation where excellent individuals are discarded by the GA. Li et al. [15] introduced evidence theory, that has good data fusion ability, since it is able to reason with uncertainty to synthesize the evidence from SRU, Gated Recurrent Unit (GRU), LSTM sources in DO, pH, TP prediction, and eventually reached a certain level of belief. The improved probability assignment function of the evidence theory, designed based on the softmax function, could solve the failure of weight allocation problems existing in the traditional probability assignment function. As a general framework of uncertain reasoning, the application of evidence theory can be further extended. Tian et al. [153] combined transfer learning (TL) and ANNs approaches which do not require a large amount of training data because TL has the ability to transfer knowledge from past tasks to predict Chl-a dynamics. The biggest difference between TL and traditional ANNs methods is that the former does not need to learn each task from scratch while the latter does. Results indicated that the hybrid models enhanced the generalization ability compared with the dropout and parameter norm penalties methods in the long-term application. At the same time, the impact of mutable data distribution on the models was decreased. Yan et al. [154] utilized mean value method using a median of k data before and after to correct wrong data and got the missing data by the values of model prediction of other water quality variables at the missing point. The restricted condition of the model was that the data were appropriately and normally distributed. Therefore, it is uncertain whether the above method can be applied to other prediction tasks that do not meet the above conditions. Yan et al. [68] proposed a hybrid optimized algorithm, namely PSO and GA, to optimize BPNN with reasonable accuracy. Y. Liu [45] investigated the DO prediction, which considered a temporal and spatial relationship. Spatial relationship refers to the spatial correlations between external variables instead of the geographic distributions. The newly proposed attention-RNN model achieved excellent performance whether in short-term and long-term prediction. Zounemat-Kermani et al. [63] tested the performance of decomposition approaches, DWT and VMD, in DO prediction. They concluded that these two methods are an alternative tool for accurate prediction when the input was combination III and model was MLP.

## 5. Result

The year of the publication is analyzed at first. Figure 7 plots the number of articles published from 2008 to 2019 each year. There is a growing number of publications since 2008 that use the ANN models to predict the water quality, including above 50% of the papers published since 2015, despite the fact that there are some fluctuations in the quantity of papers—which was in decline in 2010 and 2011. The increasing popularity of ANNs in the field of water resources [155] and environmental engineering [16] may be explained by the major advantage of the ANNs—that researchers can utilize them to model nonlinear and complex phenomena even if they do not fully understand the underlying mechanisms [156]. The popularity of ANNs above is also in agreement with the observations of other researchers [27,30]. Moreover, the number of papers for different prediction variables is summarized in Figure 8. The majority of the reviewed papers used chemical water quality variables, such as DO, BOD, and COD as outputs [30] in the systems of the river, lake, and WWTP. Furthermore, attention was also directed towards physical variables like pH, WT, and biological variables such as Chl-a.

The number of diverse forecast lengths is shown in Figure 9. The forecast length in this review refers to the length of time to predict in advance. For example, if researchers used the historical data of the previous three days to predict the values of the current day, then the forecast length would be 1 [157]. However, 107 papers did not provide details about the forecast length which cast ambiguity and doubts to researchers in parameter settings [31]. It seems ideal to utilize ANN models to capture short-term (length = 1) relationships, as the process was carried out 30 times in 44 papers which provide details about the forecast length, while only 10 papers consider long-term (length > 1) forecasting.

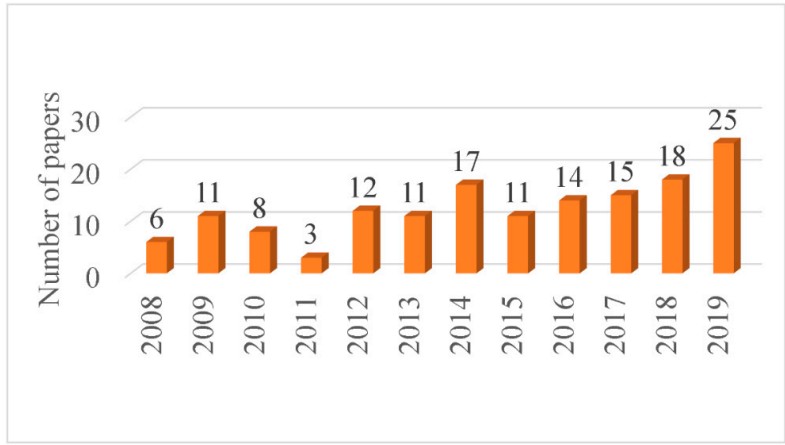

**Figure 7.** The distribution of papers between 2008 and 2019.

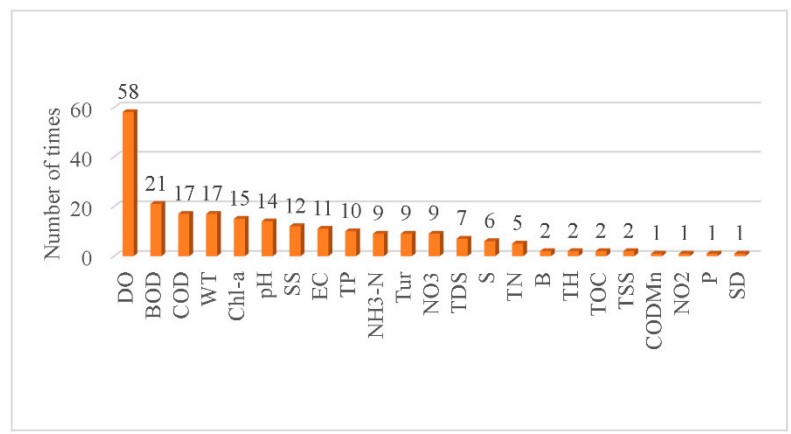

**Figure 8.** Number of papers for different prediction variables.

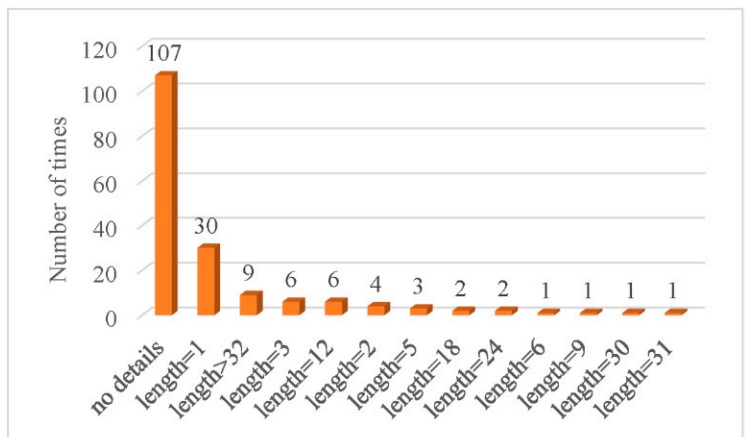

**Figure 9.** The distribution of prediction lengths.

As mentioned in the Introduction, this review not only includes more water quality parameters but also more extensive research scenarios compared with the previous reviews. On the whole, there are 23 types of water quality variables examined in this review. They are mainly physical, chemical, and biological variables. In the field of water quality prediction, relatively mature sensors include DO, WT, Chl-a, pH, EC, and NH$_3$-N. There are different application scenarios among the investigated water quality variables. Table 3 summarizes the main application scenarios of various water quality variables. Researchers conducted more prediction studies on DO, WT, Chl-a, pH, EC, NH$_3$-N, Tur, and S than other water quality variables. It can be seen from Table 3 that there are simple and practical

sensors that can measure these water quality variables. Therefore, the extensive research of the above variables may benefit from the wide application of these sensors [148].

**Table 3.** Basic information of water quality variables.

| Water Quality Variables | Categories | Unit | Major Sensors | Research Scenarios |
|---|---|---|---|---|
| DO | chemical | mg/L | ✓ | river, lake, reservoir, WWTP, ponds, coastal waters, creek, drain |
| BOD | chemical | mg/L | - | river, lake, WWTP, mine water experimental system |
| COD | chemical | mg/L | - | river, lake, reservoir, WWTP, groundwater, mine water |
| WT | physical | °C | ✓ | river, lake, ponds, catchment, stream, coastal waters |
| Chl-a | biological | µg/L | ✓ | lake, reservoir, surface water, coastal waters |
| pH | physical | none | ✓ | river, lake, WWTP, stream, coastal waters |
| SS | physical | mg/L | - | river, stream, coastal waters, creek, catchment |
| EC | physical | us cm$^{-1}$ | ✓ | river, lake, reservoir, groundwater, stream |
| TP | physical | µg/L | - | river, lake, WWTP |
| NH$_3$-N | chemical | mg/L | ✓ | river, lake, reservoir, groundwater experimental system |
| Tur | physical | FNU | ✓ | river, stream |
| NO$_3$ | chemical | mg/L | - | river, groundwater, catchment, wells, aquifer experimental system |
| TDS | physical | mg/L | - | river, groundwater, drain |
| S | physical | psu | ✓ | groundwater, coastal waters |
| TN | chemical | mg/L | - | lake, WWTP, coastal waters |
| B | physical | mg/L | - | river |
| TH | physical | mg/L | - | river |
| TOC | chemical | mg/L | - | river |
| TSS | physical | mg/L | - | river |
| COD$_{Mn}$ | chemical | mg/L | - | river |
| NO$_2$ | chemical | mg/L | - | groundwater |
| P | physical | mg/L | - | experimental system |
| SD | physical | cm | - | lake |

Table 4 summarizes the data set sizes of feedforward and recurrent neural networks involved in this review. According to Table 4, the number of samples applied for water quality prediction varies from 28 [39] to 45,594 [78] which illustrates the fact that ANN models are capable to deal with different size of the dataset. However, there has been no research studying the optimal amount of data required for each ANN model. As can be seen from Table 4, the recurrent neural networks [55] generally need more datasets compared with feedforward neural networks [139]. Research into the water quality parameter prediction have focused on rivers, WWTP, lake, and reservoir. In contrast, researchers have done little on artificial facilities, such as stream and pond. In the river system, most researchers use feed-forward neural networks for modeling, which may be due to the fact that the river system can be well analyzed using only the feed-forward neural network. This result also applies to WWTP systems. In the lake system, recurrent neural networks have shown significant results. These two kinds of neural networks have applications in reservoirs. In contrast, feed-forward neural network can predict water quality with relatively little data. In addition to being able to perform prediction tasks, GRNN is also suitable for small data sets (28, 32, 61, 151, 159, 265 samples) compared with other types of ANNs [24,39–43], so researchers should pay some attention to it.

**Table 4.** Datasets of feedforward and recurrent neural networks.

| Categories | Authors (Year) | Methods | Scenario (s) | Time Step | Dataset (Samples) |
|---|---|---|---|---|---|
| Feedforward | [39] | GRNN, BPNN, RBFNN | lake | weekly | 28 (6 months) |
| | [40] | ANN(MLP), GRNN | coastal waters | No details | 32 (5 months) |
| | [59] | BPNN | river | No details | 39 (3 days) |
| | [158] | ANN | mine water | No details | 73 |
| | [97] | ANN | groundwater | No details | 97 |
| | [106] | ANN(MLP) | surface water | No details | 110 |
| | [159] | MLP | river | No details | 110 (8 hours) |
| | [130] | ANN(MLP) | plain | No details | 122 |
| | [128] | ANN | groundwater | monthly | 124 (1 year) |
| | [80] | ANN(MLP) | stream | No details | 132 (11 months) |
| | [79] | ANN(MLP) | basin | fortnightly | 144 (1 year) |
| | [131] | ANN(MLP) | river | monthly | 144 (12 years) |
| | [121] | RBFNN | river | weekly | 144 |
| | [24] | GRNN, ANN(MLP), RBFNN, MLR | river | monthly | More than 151 samples (6 years) |
| | [42] | GRNN, MLR | Open-source data | No details | 159 (9 years) |
| | [160] | ANN(MLP) | river | monthly | 164 (over 6 years) |
| | [107] | ANN | reservoir | No details | 180 (3 years) |
| | [22] | ANN | system | No details | 195 (4 years) |
| | [161] | ANN(MLP), RBFNN | river | monthly | 200 (17 years) |
| | [139] | ANN | river | No details | 200 (16 years) |
| | [93] | ANN | river | No details | 232 (3 years) |
| | [63] | CCNN, MLP | river | half a month | 232 (12 years) |
| | [122] | ANN | river | No details | 252 (21 years) |
| | [113] | ANN(MLP) | river | No details | 255 (7 months) |
| | [43] | ANN(MLP), RBFNN, GRNN | WWTP | daily | 265 (3 years) |
| | [119] | ELM | WWTP | daily | 360 |
| | [75] | ANN | WWTP | daily | 364 (1 year) |
| | [118] | BPNN | reservoir | No details | 400 (20 years) |
| | [94] | BPNN | NA | No details | 500 |
| | [95] | ANN | river | monthly | 500 (10 years) |
| | [10] | ANN | groundwater | 30 minutes | 818 (nearly 17 days) |
| | [162] | BPNN | river | No details | 969 |
| | [77] | MLP, RBF, GRNN | Well | No details | 975 (16 years) |
| | [163] | ANN(MLP) | stream | daily | 982 (6 months) |
| | [88] | MLP | lake | No details | 1087 (6 years) |
| | [133] | ANN | lake | No details | 1217 |
| | [127] | RBFNN, GRNN, MLR | river | No details | More than 1300 samples (6 years) |
| | [36] | RBFNN, TDNN | river | monthly | 1320 (10 years) |
| | [117] | GRNN | river | No details | 1512 (9 years) |
| | [50] | MNN | WWTP | No details | 1900 |
| | [83] | ANN(MLP), RBFNN | river | daily | 2063 (6 years) |
| | [137] | ANN | river | No details | 3001 |
| | [112] | RBFNN, ANN(MLP), MLR | upstream and downstream | daily | 2063 and 4765 samples (18 years) |
| | [115] | ANN | river | daily | more than 3000 samples (11 years) |
| | [116] | ANN | lake | 15 minutes | 6674 (86 days) |
| | [164] | ANN | river | No details | 13,800 (5 years) |
| | [25] | GRNN | river | No details | more than 32,000 samples |
| | [47] | ELM, ANN(MLP) | Open-source data | hourly | 35,064 (4 years) |
| | [78] | ANN(MLP) | power station | 10 minutes | 45,594 (2 years) |

**Table 4.** *Cont.*

| Categories | Authors (Year) | Methods | Scenario (s) | Time Step | Dataset (Samples) |
|---|---|---|---|---|---|
| | [41] | Elman, GRNN, BPNN, MLR | river | monthly or semi-monthly | 61 |
| | [12] | NARX, BPNN, MLR | river | monthly | 280 (11 years) |
| | [26] | LSTM | river | 12 hours | 460 (14months) |
| | [6] | LSTM, BPNN | lake | monthly | 657 (7 years) |
| | [66] | LSTM, RNN | Mariculture base | 5 minutes | 710 (21 days) |
| Recurrent | [67] | SRU | Mariculture base | No details | 710 |
| | [3] | Elman | pond | No details | 816 (34 days) |
| | [153] | RNN, LSTM | reservoir | 5 minutes | 1440 (5 days) |
| | [15] | RNN, BPNN | river | No details | 1448 |
| | [165] | LSTM | lake | No details | 1520 |
| | [54] | LSTM, BPNN | pond | 10 minutes | 2880 (20 days) |
| | [38] | RESN | WWTP | No details | 5000 |
| | [45] | RNN | Freshwater | 10 minutes | 5006 (1 year) |
| | [55] | TLRN, RNN, TDNN | lake | 15 minutes | 13,744 (573 days) |
| | [154] | LSTM | WWTP | hourly | 23,268 (4 years) |

The artificial neural network has been widely used in water quality prediction. If researchers only look at the modeling process, various studies follow some of the steps of the modeling framework below (see Figure 10).

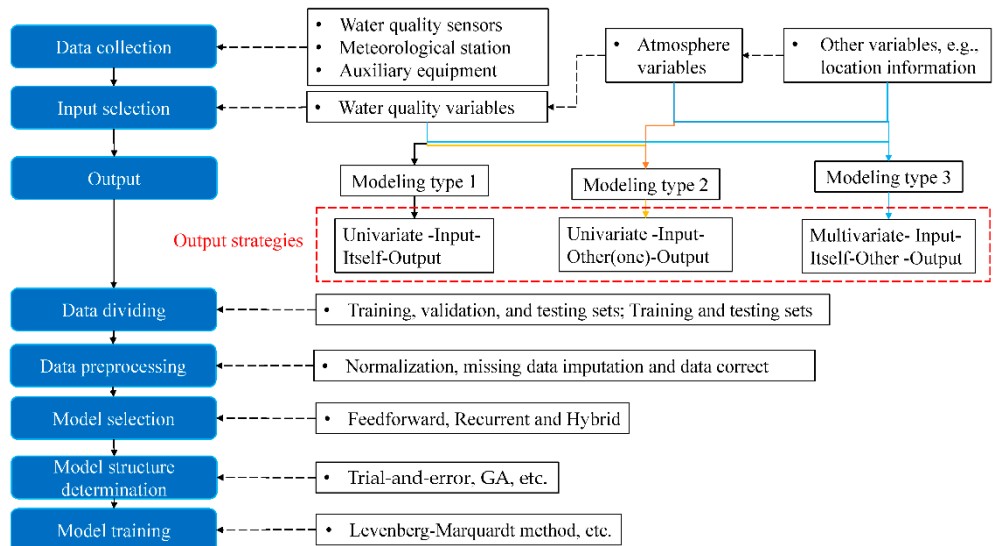

**Figure 10.** General framework for water quality modeling.

## 5.1. Data Collection

The data collection process is not easy due to the requirement of costly measuring instruments (e.g., water quality sensors, meteorological stations), laboratory equipment, and good operating conditions. Water quality variables are primarily collected by the sensors. Meteorological variables, such as AT, WS, RF, SR, Precip, and AP, often influence water quality. Therefore, some researchers took the meteorological station to obtain the data. In addition, some parameters, such as BOD, COD, need to be measured by auxiliary laboratory equipment [44]. Location information is essential when researchers want to make a three-dimensional prediction of water quality. In the above case, the required data is obtained through the device (see Figure 10). In some studies [42,47], the researchers conduct studies based on an open-source dataset.

Based on the obtained data, researchers can perform three modeling types. The first type of modeling is where the researcher models only historical information about the output variable.

The second type of modeling is when the output variables are difficult to measure, and the researchers can use easily measured water quality or meteorological data to complete the prediction. In the first two modeling types, the researchers utilized univariate historical information. However, for the third type of modeling, the researchers used multivariable historical information. Overall, the researchers utilized water quality, atmosphere, and other variables such as location data for the prediction task. The above three modeling types are analyzed from the perspective of data. If analyzed from the perspective of studying the temporal and spatial relationship between input and output, the above modeling types can be further divided.

*5.2. Output Strategy*

The output strategies can be further divided into five categories based on the three modeling types (see Figure 10). Temporal relationship refers to the relationship learning in the time dimension. Spatial relationship [45] refers to the spatial correlations between external variables (see Figure 11). The black origin describes a variety of input variables. Table 5 summarized the detailed descriptions of the five output strategies. Simply speaking, external variables are the other variables (more than one) in Multivariate-Input-Itself-Other(multi)-Output. Univariate-Input-Itself-Output [64] and Univariate-Input-Other(one)-Output [79] refer to the univariate case, while Multivariate-Input-Other (multi)-Output [35], Multivariate-Input-Itself-Other-Output [52], and Multivariate-Input-Itself-Other (multi)-Output are multivariate [45] (see Table 5). The model learns the temporal relationship from five output strategies, while the spatial relationship is only considered in Multivariate-Input-Itself-Other (multi)-Output. The distinctions between Univariate-Input-Other (one)-Output and Multivariate-Input-Other (multi)-Output are not only the number of input variables, but also the fact that the former's output strategy focuses on time series data while the latter contains more. The main difference between Multivariate-Input-Itself-Other-Output and Multivariate-Input-Other (multi)-Output is that the former uses the historical information of the output variable, while the latter does not.

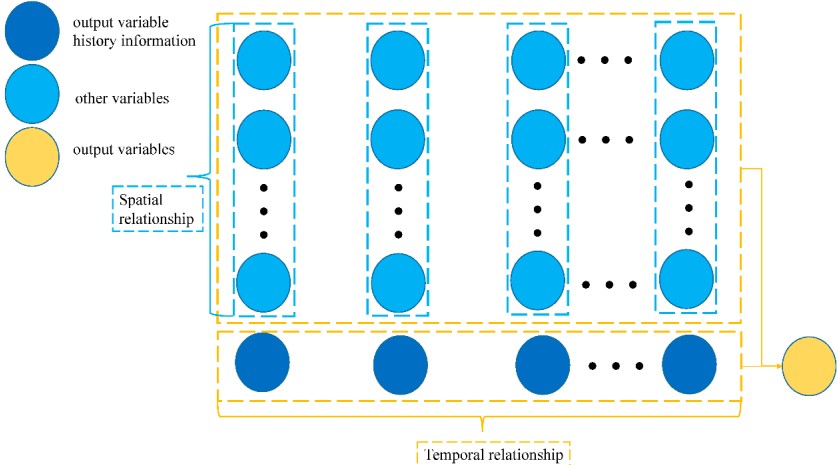

**Figure 11.** Temporal and spatial relationship in Multivariate-Input-Itself-Other (multi)-Output.

**Table 5.** Five different output strategies.

| Category | Type | Relationship | Description |
|---|---|---|---|
| Univariate-Input-Itself-Output (Category 0) | Univariate | Temporal relationship | The output(s) at a specific point are learned from its own historical information |
| Univariate-Input-Other(one)-Output (Category 1) | Univariate | Temporal relationship | The output(s) at a specific point are learned the historical information from other variables (one) |
| Multivariate- Input-Other (multi)-Output (Category 2) | Multivariate | Temporal relationship | The output(s) at a specific point are learned the historical information from other variables (more than one) |
| Multivariate-Input-Itself-Other-Output (Category 3) | Multivariate | Temporal relationship | The output(s) at a specific point are learned the historical information from both its own and other variables |
| Multivariate-Input-Itself-Other (multi)-Output (Category 4) | Multivariate | Temporal relationship and spatial relationship | The output(s) at a specific point are learned the historical information from both its own and other variables (more than one) |

## 5.3. Input Selection

There are two main approaches to select the most significant predictors of ANN models which are model-free and model-based methods (see Table 6) [166]. The biggest difference between the two methods is that the former does not consider model performance, while the latter does. In the majority of the studies, many researchers utilized ad-hoc [27] methods to select the inputs, whether in model-free or model-based methods. Some researchers used cross-correlation and analytical approaches to explore the linear and non-linear relationship between input(s) and output(s). Other input selection methods are summarized in Table 6.

**Table 6.** Model-free and model-based methods in input selection.

| Categories | Methods | Comments |
|---|---|---|
| model-free | ad-hoc<br>analytic<br>other | Based on domain knowledge or casual way<br>The linear and non-linear relationship between input and output<br>IGRA, Garson method |
| model-based | ad-hoc<br>stepwise<br>sensitivity analysis<br>global optimization | e.g., trial-and-error<br>Constructive and pruning methods<br>e.g., MCS<br>e.g., GA |

## 5.4. Data Dividing

Data dividing is an important step in the modeling process (see Table 7). The training set is used for data samples of model fitting [95]. The validation set, which can adjust the model's hyperparameters, is a set of samples set apart during model training. Finally, the testing set is to check the model's generalization ability [139] and its error is utilized to compare different model's predictive performance. Not all data needs to be divided into three sets, because regularization [55] is an approach that can divide the datasets into two sets—namely training and validation sets—and has the advantage of providing more data points for the model training and stopping the models from over-fitting [167]. Data dividing methods can be categorized into supervised and unsupervised methods [31]. There are

no uniform rules for how to divide the training set, the validation set, and the test set which also applies to the division of training sets and test sets. Most researchers divided the data either by domain knowledge or in any arbitrary manner. In the majority of the reviewed papers, the data set was divided into the training and testing two parts (see the ninth column in Appendix A). The division range of the training set is from 50% to 98% [16], and the test set varies from 2% to 50% [105]

**Table 7.** Supervised and unsupervised methods in data dividing.

| Categories | Methods | Comments |
|---|---|---|
| supervised | trial-and-error<br>temporal partitioning<br>M-test | Taking the statistical properties of each subset into consideration<br>Dividing the data into diel, diurnal, and nocturnal<br>The number of the data points was obtained through the winGamma software |
| unsupervised | ad-hoc<br>random<br>cross-validation<br>stratified method | Based on domain knowledge or a casual way<br>Divide the data randomly<br>e.g., K-fold cross-validation, leave-one-out cross-validation<br>e.g., SOM |

*5.5. Data Preprocessing*

It should be noted that data preprocessing is carried out after the data dividing. Normalization, missing values imputation and data correct are three primary preprocessing methods in the field of water quality modeling (see Table 8). Most reviewed papers took the normalization step, although they did not give clear details about normalization. As [31] pointed out, this step requires matching the range of the predictors to the transfer function in the hidden layer. Range scaling [132] and standardization [113] are two popular categories in normalization. There are three main scopes, namely [0, 1], [−1, 1] and [0.1, 0.9], under range scaling. Although missing data often occurs in transmission, only a few investigated papers dealt with this phenomenon. The majority of researchers deleted the missing data directly. This is not a recommended practice, as the obtained data are precious and limited. As a whole, researchers pay less attention to data imputation, correct, and identification of abnormal data. Table 4 presented some data preprocessing techniques.

**Table 8.** The data preprocessing approaches.

| Categories | Methods | Comments |
|---|---|---|
| Normalization | No details<br>Range scaling<br>Standardization | Built-in functions in platforms<br>The scale of each feature is in the same range<br>A new variable with zero mean and unit standard deviation |
| Missing data imputation | Only mentioned<br>Deletion<br>Linear interpolation<br>Improved mean value method<br>Missing–refilling scheme<br>Gap-filling<br>Filling in the predicted values of the model | Not recommend<br>Not recommend<br>The slope of the assumed line to calculate the data increment<br>Solve the breakpoint phenomenon of mean value method and linear interpolation method<br>Dividing of ID and SD and using Temporal exponentially moving average to fill the missing data<br>Temporal partitioning as gap-filling in order to get continuous records<br>The missing values of predictors at time T0 are obtained by prediction values of the model at time T0 by other predictors |
| Data correct | Smoothing method<br>Mean value method | The moving average filtering can attenuate high-frequency signals<br>Need to be corrected as a median of k data before and after |
| Data abnormal | The fixed threshold method | Setting the upper and lower threshold ranges (discard) |

*5.6. Model Structure Determination*

Until recently, a general method for determining the optimal model structure remains unknown [31]. Therefore, different approaches have been adopted to determine the ANN model structure to avoid the initial difficulty in model building step as much as possible. There are three mainstream

methods—namely ad-hoc, stepwise trial-and-error, and global methods—for the determination of an optimal model structure [31] (see Table 9). The neural network structure defines the functional form of the input–output relationship [59]. The model structure determination, an essential step in the model development, refers to the number of layers, the number of nodes in each layer and the way they connect [30], aiming to strike a balance between network complexity and generalization ability [27]. The model structure determination and model training process are often conducted together. For example, when the trial-and-error method is implemented, the weights of the MLPs are optimized at the same time. Categories and comments on the ANN methods in the model structure determination are given in Table 9. M, N, and O are the number of neurons of the hidden layer, input layer, and output layer. A is a constant from 1 to 10. Sqrt is a mathematical function [83] used to calculate the square root of a non-negative real number. Nearly half of the investigated papers did not provide details on the methods used to determine the ANNs structure. When using fixed network structures such as GRNNs, this step is not necessary to carry out, although its proportion is relatively small compared with papers that did not mention this step (see the last two lines in Table 9). In 73 of the 90 times which provide details of the methods, ad-hoc approaches were utilized to determine the structure of model. That is to say, most studies still rely on trial-and-error approach to determine the model structure. This also reveals that researchers have not been very innovative in the methodology of model structure determination. Seven empirical formulas can help to determine the structure of the model to a certain extent in the investigated articles. Table 9 also presents the various global approaches and their improvements in the reviewed articles.

**Table 9.** Three main model structure determination methods.

| Categories | Methods | Comments | Typical Examples |
|---|---|---|---|
| Ad-hoc | Empirical formula and trial-and-error approach | Rule 1: M is less than N minus 1 | |
| | | Rule 2: one range of M is equal to the sqrt of N plus O and finally plus A | [123] |
| | | Rule 3: the other range of M is equal to log base 2 logarithm of N | |
| | | Rule 4: M is equal to 5 multiplied by sqrt of N | [102] |
| | | Rule 5: M is equal to half of the sum of N and O plus square root of the number of training patterns | [102] |
| | | Rule 6: M is equal to sqrt of N plus one and finally plus A | [33] |
| | | Rule 7: M is equal to sqrt of N multiplied by O | [99] |
| | Trial-and-error | Purely on a trial-and-error approach | [105] |
| Stepwise trial-and-error | Stepwise trial-and-error | With each modification of the trial, a structure that is neither too complex nor too simple is building | [99] |
| Global methods | GA | Searching the solution space through simulated natural evolution | [166] |
| | u GA | Introducing creep mutation in a small population | [140] |
| | IGA | Selecting excellent individuals effectively to avoid the situation of discarding by GA | [152] |
| | PSO | Excitation function does not need to be differentiable and derivable | [143] |
| | IPSO | The convergence rate and accuracy of the solution are improved | [148] |
| | ABC | More precise than PSO and GA | [4] |
| | IABC | Updating formulas just like the PSO algorithm | [4] |
| Others | Not mentioned | Not recommend | [40] |
| | Not required | Fixed structures such as GRNNs | [25] |

### 5.7. Model Training

There are two main training methods, namely deterministic and stochastic methods [31] (see Table 10). Deterministic methods look for a single parameter vector while the stochastic methods search for the distribution of the model parameters with the purpose of minimizing the model error [27]. In a more detailed division, local methods (L) that often work on gradient information and global optimization approaches are two kinds of deterministic methods. Gradient methods can be further sub-divided into first-order methods or second-order methods. Deterministic methods based on gradient information have been widely used in model training algorithm. The Levenberg–Marquardt algorithm, a second-order method, was most widely used in deterministic methods. The Levenberg–Marquardt method combines the advantages of BP and Newton algorithm, and its training speed is obviously faster than BP and momentum algorithm [81]. However, it has the disadvantage of being incompatible with regular terms, and requires a lot of memory when datasets are large. Sixty-two of the papers did not provide details about the model training algorithm. Seven categories of local methods (see line 1 to line 7 in Table 10) are summarized. Relatively speaking, there were few works on network training using Bayesian [27] and the Adam optimization methods [57].

**Table 10.** The deterministic and stochastic methods in model training.

| Categories | Methods | Comments |
|---|---|---|
| Deterministic | BP algorithm(L) | Computing the direction of gradient descent |
| | Newton's methods(L) | The computing tasks are implemented by Hessian matrix |
| | Conjugate gradient method(L) | The search direction is carried along the conjugate direction and does not need to use Hessian matrix |
| | Levenberg–Marquardt method(L) | A method, combination of BP and Newton algorithm, use Jacobian matrix to do the computing tasks |
| | The Quasi-Newton method(L) | It is applied to the situation of that Jacobian matrix or Hessian matrix is difficult or even impossible to compute |
| | BFGS | A Quasi-Newton method implemented by the built-in function in R |
| | TRAINLM | A gradient descent with momentum and Levenberg–Marquardt backpropagation |
| | Global optimization | See Table 9 |
| Stochastic methods | Bayesian methods | Prediction limits can be obtained |
| | Adam optimization method | It implemented a reverse gradient update with the value obtained by Mini batch data |
| Emerging methods | Online learning algorithm | Quickly adjust the model in real time |

## 6. Discussion

### 6.1. Data Are the Foundation

Data selection strategy: Data collection is a costly and time-consuming process. This is mainly due to the expensive equipment, limited experimental time, and conditions. The ANN model is a data-driven model, so obtaining enough data is the basis of modeling. The need to collect as much data as possible has been put forward in the existing literature. To address this need, researchers need to consider two factors. One is whether the historical information of the output variable can be collected. The second is what strategy researchers need to choose when the historical information of output variables are difficult to measure. If researchers can collect historical observations of target variables, they can process the data and model it. If target variables cannot be obtained, researchers can collect variables such as water quality and meteorology data associated with output variables. Part of the literatures collect target variables by means of obtaining open-source data. This approach has benefited from a number of government data collection programs. However, research to obtain open-source

water quality data is rather limited. Therefore, researchers are encouraged to open up their own data resources in the future to make contributions to themselves, others, and society.

Data volume demand: According to the results of the existing literature review, there is no systematic research to investigate how to determine the optimal number of samples required for each type of ANN model. In general, RNN requires more data than feedforward artificial neural networks. In addition, GRNN in feedforward artificial neural networks can handle small sample problems. When researchers use the RNN method to make water quality predictions, they need about a thousand pieces of data. When the researchers utilized the feedforward artificial neural networks method to make the prediction, about 500 pieces of data are needed. When researchers used the GRNN method to make predictions, they need about 100 pieces of data. When researchers want to make long-term forecasts of water quality data that are periodic after a year, at least one year of data needs to be collected. This also applies when researchers want to include four seasons in the model validation and testing phase.

*6.2. Data Processing Is Key*

Data imbalance problem: Both the peak and the extreme value occupy a relatively small proportion of the distribution. Only a handful of researchers currently consider data imbalances. In order to obtain higher prediction accuracy and reduce the error of the peak, some new prediction approaches, such as wavelet analysis method, can be used for reference. Besides, modelers in the future can develop a form of extreme value loss for detecting the future occurrence of extreme values (Ding et al., 2019) and apply it to the water quality prediction.

Input selection problem: The quality of data sets has been affected by many factors. These factors include but are not limited to temporal resolution (e.g., monthly vs. hourly), number of predictors, or noise in the data. Therefore, it is very important to select the appropriate input and preprocess the data. This review found that the vast majority of researchers chose inputs based on their domain knowledge or in any arbitrary manner. Such input selection methods have some limitations because they neither analyze the relationship between input and output, nor consider the performance of the model. Some studies use cross-correlation to explore the relationship between inputs and outputs. It is a linear approach, which is contrary to the premise of using a nonlinear neural network model. Researchers can use nonlinear analysis methods such as mutual information to select inputs.

Output strategies problem: A variety of output strategies were adopted in the reviewed papers—the quantity of which is 18—because researchers hope to select the most suitable through comparison to illustrate the relationship between input(s) and output(s), which is good practice. Multivariate-Input-Other (multi)-Output is the most popular output strategy which represents the case where the output(s) at a specific point is learned the historical information from other variables (more than one). Few studies have considered the spatial relationship between exogenous variables. This may be due to the fact that external variables do not influence the outcome of the forecast most of the time. However, researchers must be aware that exogenous variables can have a significant impact on predictions at some point. For example, the effect of water circulation on dissolved oxygen. A recent research used the mechanism of attention to simultaneously explore the relationship between temporal and spatical, and applied it to DO prediction. Researchers can use this method for reference to further explore the spatial relationship of other water quality variables.

Forecasting length problem: At present, the research mainly focuses on the short-term prediction, and the research on the long-term prediction is relatively limited. The reason for this phenomenon is that with the increase in the prediction length, the uncertainty factors also increase, which leads to the accumulation of errors and thus reduces the accuracy of the prediction. Researchers can adopt appropriate strategies to solve such problems in forecating field, such as Recursive, DirRec, and Multiple Output Strategies [168].

Data dividing problem: At present, researchers tend to use ad-hoc method to divide the training set data into 70% to 90% of the total data. The most common percentage of the training, validation,

and testing is 60%, 20%, 20%, and 50%, 25%, 25%. Such methods based on the expertise of researchers or divide data in arbitrary ways has certain universality. However, this approach has not promoted the development of data partitioning methods. It is always difficult to determine the number of K for common K-fold cross-validation, as the results may have a considerable bias [169]. Therefore, leave-one-out cross-validation, the most extreme form of K-fold cross-validation, should be encouraged for use because it has been shown to provide a good estimation of the model's true generalization capabilities in the case of fewer training data or more model parameters despite the limited usage.

Data preprocessing problem: Most studies use the normalization method for pre-processing data, but it does not disclose specific details. This is probably due to the use of built-in functions to deal with normalization in many platforms. However, this basic information should be clearly defined, because different scaling ranges have different effects on the final result of the model. In the face of missing data, researchers will simply delete it. This approach is not worth advocating because data is precious. Researchers can adopt appropriate populating strategies to deal with missing data. Some imputation methods besides linear interpolation—such as the improved mean value method that can solve the breakpoint phenomenon of linear interpolation, and designing filling schemes such as missing–refilling schemes or gap-filling to obtain continuous records—are worthy of exploring. The restricted condition of the model forecasting methods using prediction values to fill the missing vales is that the data are appropriately and normally distributed. Therefore, it is uncertain whether the above method can be applied to other prediction tasks that do not meet the above conditions. Existing literature has shown that the identification of error and abnormal data is a difficult task because they are difficult to define in water quality prediction. How to deal with such data still needs further exploration by researchers.

### 6.3. Model Is the Core

Model structure determination problem: Most researchers use a trial-and-error method to determine the ANN structure, which does not fundamentally promote the further development of the model. This review summarizes some empirical formulas to determine the number of neurons in the hidden layer that future researchers can apply to their studies. This review does not reveal the science behind these formulas or the conditions under which they apply. To some extent, the use of these empirical formulas contributes to the determination of the model structure, because researchers build on previous studies rather than stay at the level of trial-and-error with no rules to find. Global methods can obtain topology and network weights, which have been developed to some extent in recent years. Compared with the trial-and-error method, the global method has a sound theoretical basis. Researchers can further study and improve global methods.

Activation function determination problem: Most of the time, researchers choose S-shaped functions because they create a random nonlinear mapping between the input and output. The essential reason is that the S-type transfer function is differentiable, continuous, and monotonically increasing in its domain. Purelin is used more frequently in the output layer than other functions because its output can be arbitrary rather than limited to a small range compared to the sigmoid function.

Model training problem: The reason for developing so many subclasses of training algoriths is that researchers want to use the appropriate matrix (e.g., Hessian matrix, Jacobian matrix) to accomplish the computing tasks easily. The Quasi-Newton method is suited for the situation that the matrix (whether Hessian or Jacobian) is difficult or even impossible to compute. In water quality prediction, the deterministic methods are more mature than the stochastic methods. One possible reason is that the former only looks for a single parameter vector, while the latter looks for the model parameter distribution, so the latter parameter is more uncertain. The online learning algorithm has the characteristics of real-time and rapid adjustment model which is suitable for prediction tasks. However, its application in water quality prediction is still very limited. Therefore, the algorithm is worthy of further study.

Model structure selection problem: Many researchers utilized MLP architectures in ANN to complete prediction tasks between 2008–2019. This result is as same as the conclusion of the review between 1999 to 2007, which may be due to the fact that the MLPs architecture has the advantage of being easy to use, and they can approximate any relationship between input(s) and output(s) through the typical three layers [81]. Global methods (see Table 9), obtaining topology structure and network weights, are drawing the attention of researchers—in contrast to the previous review [27]. It must be noted that the GA, PSO, and ABC methods are typical examples of evolution-related methods. In general, evolutionary methods are combined with ANNs to meet different constraints.

Much effort has been made regarding the data-intensive methods, while the model-intensive and technique-intensive approaches were implemented relatively infrequently. Wavelet analyses were widely used in data-intensive methods, while the decomposing approaches were used less. This may be because wavelet analysis has the ability to extract the trends, discontinuities, and breakdown points of the original data. Furthermore, it is also able to process signals by compressing or denoising.

In recent years, CNN, as a new feedforward neural network method, has been used in water quality prediction. However, its application is rather limited. Researchers can further expand CNN's reach. RNN has good memory ability, so it can make full use of historical information and lay a solid foundation for realizing long-term prediction of water quality. Hybrid Models should be further developed because they are not a substitute for traditional technologies, but a combination of their strengths. Researchers can refer to the ensemble approaches, transfer learning technology, and evidence theory in the literature to improve the prediction accuracy and generalization ability, and accommodate uncertainty.

**Author Contributions:** Conceptualization and review framework, Y.C.; original draft preparation and writing, L.S.; review and editing, Y.L., L.Y.; supervision, D.L. All authors have read and agreed to the published version of the manuscript.

**Funding:** This research was funded by the Next Generation Precision Aquaculture: R&D on intelligent measurement, control technology (Project Number:2017YFE0122 100).

**Acknowledgments:** The authors would like to thank the anonymous reviewers for their valuable and insightful comments on an earlier version of this manuscript.).

**Conflicts of Interest:** The authors declare no conflict of interest.

## Appendix A

**Table A1.** Details of the reviewed papers.

| Categories | Authors (Year) | Locations | Water Quality Variables | Meteorological Factors | Other Factors | Output Strategy | Dataset | Time Step | Data Dividing | Methods | Prediction Lengths |
|---|---|---|---|---|---|---|---|---|---|---|---|
| Feedforward | [75] | WWTP(Turkey) | BOD; SS, TN, TP | NA | Q | Category 2 | 364 samples (1 year) | daily | Train: 67%, test:33% | ANN, MLR | NA |
| Feedforward | [76] | Mamasin dam reservoir (Turkey) | DO, EC; SS, TN, WT | RF | AODD | Category 2 | No details | No details | No details | ANN(MLP) | NA |
| Feedforward | [40] | Singapore coastal waters (Singapore) | S, DO, Chl-a;; WT | NA | NA | Category 3 | 32 samples (5 months) | No details | No details | ANN(MLP), GRNN | 1 |
| Feedforward | [19] | Feitsui Reservoir (China) | Chl-a; | NA | Bands | Category 2 | No details | No details | Train: 75%, test:25% | ANN(MLP) | NA |
| Feedforward | [90] | Pyeongchang river (Korea) | DO, TOC; WT | NA | Q | Category 3 | No details (3 months) | 5 minutes | No details | ANN, MNN, ANFIS | 12,24 |
| Feedforward | [170] | Feitsui Reservoir (China) | Chl-a; | NA | Bands | Category 2 | No details (7 years) | No details | Train:57%, validate: 29%, test: 14% | ANN(MLP) | NA |
| Feedforward | [91] | Melen River (Turkey) | BOD; COD, WT, DO, Chl-a, NH$_3$-N, NO$_3$, NO$_2$ | NA | F, Ns | Category 2 | No details (over 6 years) | monthly | Train:60%, validate: 20%, test: 20% | ANN(MLP) | NA |
| Feedforward | [92] | Moshui River (China) | COD, NH$_3$-N;; | NA | mineral oil;; | Category 0 | No details (5 years) | No details | Train:80%, test: 20% | BPNN | NA |
| Feedforward | [93] | Doce River (Brazil) | WT, pH, EC, TN | NA | other ions | Category 2 | 232samples (3 years) | No details | Train:50%, validate: 25%, test: 25% | ANN | NA |
| Feedforward | [94] | NA (China) | pH, DO;; WT, S, NH$_3$-N, NO$_2$ | NA | NA | Category 3 | 500 samples | No details | Train:80%, test: 20% | BPNN | NA |
| Feedforward | [95] | Gomti river (India) | DO, BOD; pH, TA, TH, TS, COD, NH$_3$-N, NO$_3$, P | RF | NA | Category 2 | 500 samples (10 years) | monthly | Train:60%, validate: 20%, test: 20% | ANN | NA |

**Table A1.** *Cont.*

| Categories | Authors (Year) | Locations | Water Quality Variables | Meteorological Factors | Other Factors | Output Strategy | Dataset | Time Step | Data Dividing | Methods | Prediction Lengths |
|---|---|---|---|---|---|---|---|---|---|---|---|
| Feedforward | [96] | Pyeongchang River (Korea) | TOC;; | Precip | Q;; | Category 3 | No details (7 years) | No details | No details | ANN | NA |
| Feedforward | [97] | Groundwater (China) | $NO_2$, COD;; | NA | other 7 variables | Category 3 | 97 samples | No details | Train:56%, test: 44% | ANN | NA |
| Feedforward | [98] | Omerli Lake (Turkey) | DO; BOD, $NH_3$-N, $NO_3$, $NO_2$, P | NA | NA | Category 2 | No details (17 years) | No details | No details | ANN, MLR, NLR | NA |
| Feedforward | [99] | Changle River (China) | DO, TN, TP;; WT | RF | F, FTT | Category 3 | No details (18months) | monthly | No details | BPNN | NA |
| Feedforward | [105] | Sangamon River (USA) | $NO_3$;; | AT, Precip | Q | Category 3 | No details (6 years) | weekly | Train:50%, test: 50% | ANN | 1 |
| Feedforward | [106] | Surface water (Turkey) | Chl-a; | NA | other 12 variables | Category 2 | 110 samples | No details | Train:67%, test: 33% | ANN(MLP) | NA |
| Feedforward | [107] | Gruˇza reservoir (Serbia) | DO; pH, WT, CL, TP, $NO_2$, $NH_3$-N, EC | NA | Fe, Mn | Category 2 | 180samples (3 years) | No details | Train:84%, test: 16% | ANN | NA |
| Feedforward | [108] | The tank (China) | DO;; pH, S, WT | AT | NA | Category 3 | No details (22 months) | 1 minute | Train:57%, validate: 29%, test: 14% | ANN | 30 |
| Feedforward | [109] | Groundwater (India) | S; EC | NA | WL, T, Pumping, Rainp | Category 2 | No details (7 years) | No details | Train:29%, test: 71% | ANN | NA |
| Feedforward | [110] | WWTP(China) | BOD; COD, SS, pH, $NH_3$–N | NA | Oil | Category 2 | No details | No details | Train:50%, test: 50% | RBFNN | 5 |
| Feedforward | [10] | Groundwater (Iran) | $NO_3$; pH, EC, TDS, TH | NA | Mg, Cl, Na, K, $HCO_3$, $SO_4$, Ca, ICs | Category 2 | 818samples (nearly 17days) | 30 minutes | Train:70%, test: 30% | ANN, Linear regression (LR) | NA |
| Feedforward | [77] | Wells (Palestine) | $NO_3$; | NA | Q, other five variables | Category 2 | 975samples (16 years) | No details | No details | MLP, RBF, GRNN | NA |
| Feedforward | [112] | Upstream and downstream (USA) | DO; pH, WT, EC | NA | Q | Category 2 | 2063, 4765 samples (18 years) | daily | Train:50%, validate:25%, test: 25% | RBFNN, ANN(MLP), MLR, | NA |
| Feedforward | [50] | WWTP (Korea) | DO;; $NH_3$-N | NA | NA | Category 3 | 1900 samples | No details | Train:45%, validate:5%, test: 50% | MNN | NA |

**Table A1.** *Cont.*

| Categories | Authors (Year) | Locations | Water Quality Variables | Meteorological Factors | Other Factors | Output Strategy | Dataset | Time Step | Data Dividing | Methods | Prediction Lengths |
|---|---|---|---|---|---|---|---|---|---|---|---|
| Feedforward | [79] | Eastern Black Sea Basin (Turkey) | SS; Tur | NA | NA | Category 1 | 144 samples (1 year) | fortnightly | Train:75%, validate:8%, test: 17% | ANN(MLP) | NA |
| Feedforward | [113] | Kinta River (Malaysia) | DO, BOD, $NH_3$-N, pH, COD, Tur;; | NA | NA | Category 2 | 255 samples (7 months) | No details | Train:80%, validate:10%, test: 10% | ANN(MLP) | NA |
| Feedforward | [78] | Power station (New Zealand) | WT; | AT, AP, WD, WS | other 8 variables | Category 2 | 45,594 samples (2 years) | 10 minutes | Train:70%, test: 30% | ANN(MLP) | 12 |
| Feedforward | [114] | Yuan-Yang Lake (China) | WT; | SR, AP, RH, AT, WS, WD | ST | Category 2 | No details (2 months) | 10 minutes | Train:70%, validate & test: 30% | ANN(MLP) | 1 |
| Feedforward | [22] | Experimental system (UK) | BOD, $NH_3$-N, $NO_3$, P; DO, WT, pH, EC, TSS, Tur | NA | RP | Category 2 | 195samples (4 years) | No details | Train: 62%, test: 38% | ANN | NA |
| Feedforward | [11] | Lake Fuxian (China) | DO, TP, SD, Chl-a;; TN, WT, pH | NA | Month; | Category 2 and Category 3 | No details | No details | No details | ANN | NA |
| Feedforward | [115] | Doiraj River (Iran) | SS; | RF | Q | Category 1 and Category 2 | more than 3000 samples (11 years) | daily | No details | ANN, Support vector regression (SVR) | 1 |
| Feedforward | [116] | Lake Abant (Turkey) | DO, Chl-a; WT, EC | NA | MDHM | Category 2 | 6674 samples (86 days) | 15 minutes | Train:60%, validate:15%, test: 25% | ANN, Multiple nonlinear regression (MNLR) | NA |
| Feedforward | [37] | Johor River, Sayong River (Malaysia) | TDS, EC, Tur; | NA | NA | Category 1 | No details (5 years) | No details | The test set is approximately 10–40 % of the size of the training data set | ANN(MLP), RBFNN, LR | NA |
| Feedforward | [158] | Mine water (India) | BOD, COD; WT, pH, DO, TSS | NA | other | Category 2 | 73 samples | No details | Train:79%, test: 21% | ANN | NA |

Table A1. *Cont*.

| Categories | Authors (Year) | Locations | Water Quality Variables | Meteorological Factors | Other Factors | Output Strategy | Dataset | Time Step | Data Dividing | Methods | Prediction Lengths |
|---|---|---|---|---|---|---|---|---|---|---|---|
| Feedforward | [160] | Heihe River (China) | DO; pH, $NO_3$, NH3-N, EC, TA, TH | NA | Cl, Ca | Category 2 | 164 samples (over 6 years) | monthly | Train:60%, validate:20%, test: 20% | ANN(MLP) | NA |
| Feedforward | [117] | Danube River (Serbia) | DO; WT, pH, $NO_3$, EC | | Na, CL, $SO_4$, $HCO_3$, other 11 variables | Category 2 | 1512 samples (9 years) | No details | Train:70%, validate:20%, test: 10% | GRNN | NA |
| Feedforward | [80] | Stream Harsit (Turkey) | SS; Tur | NA | TCC, TIC | Category 1 and Category 2 | 132 samples (11months) | No details | No details | ANN(MLP) | NA |
| Feedforward | [118] | Feitsui Reservoir (China) | DO; WT, pH, EC, Tur, SS, TH, TA, $NH_3$-N | NA | NA | Category 2 | 400 samples (20 years) | No details | No details | BPNN, ANFIS, MLR | NA |
| Feedforward | [163] | Stream (USA) | WT; | AT | Form attributes, forested land cover | Category 2 | 982 (6 months) | daily | Train:90%, validate & test: 10% | ANN(MLP) | NA |
| Feedforward | [49] | The Bahr Hadus drain (Egypt) | DO, TDS;; | NA | NA | Category 0 | No details | monthly | Train:80%, test: 20% | CCNN, BPNN | NA |
| Feedforward | [161] | Karoon River (Iran) | DO, COD, BOD; EC, pH, Tur, $NO_3$, $NO_2$, P | NA | Ca, Mg, Na | Category 2 | 200 samples (17 years) | monthly | Train:80%, test: 20% | ANN(MLP), RBFNN, ANFIS | NA |
| Feedforward | [121] | Manawatu River (New Zealand) | $NO_3$; | NA | EMS (Energy, Mean, Skewness) | Category 1 | 144 samples | weekly | Train: 70%, test: 30% | RBFNN | NA |
| Feedforward | [119] | WWTP (China) | BOD; DO, pH, SS | NA | F, TNs | Category 2 | 360 samples | daily | Train: 83%, test: 17% | HELM, Bayesian approach, ELM | NA |
| Feedforward | [35] | Nalón river (Spain) | Tur; $NH_3$-N, EC, DO, pH, WT | NA | NA | Category 2 | No details (1 year) | 15 minutes | Train: 90%, test: 10% | ANN(MLP) | NA |
| Feedforward | [128] | Groundwater (Turkey) | pH, TDS, TH | NA | SAR, SO4; CL | Category 2 | 124 samples (1 year) | monthly | Train: 84.1%, test: 15.9% | ANN | NA |

**Table A1.** *Cont.*

| Categories | Authors (Year) | Locations | Water Quality Variables | Meteorological Factors | Other Factors | Output Strategy | Dataset | Time Step | Data Dividing | Methods | Prediction Lengths |
|---|---|---|---|---|---|---|---|---|---|---|---|
| Feedforward | [89] | Johor River (Malaysia) | DO; WT, pH, $NO_3$, $NH_3$-N | NA | NA | Category 2 | No details (10 year) | monthly | Train:60%, validate: 25%, test: 15% | ANN(MLP), ANFIS | NA |
| Feedforward | [129] | The Taipei Water Source Domain (China) | Tur; | RF | NA | Category 2 | No details (1 year) | No details | No details | BPNN | NA |
| Feedforward | [130] | Mashhad plain (Iran) | EC; | NA | CL; Lon, Lat | Category 2 and Category 3 | 122 samples | No details | Train:65%, validate: 20%, test: 15% | ANN(MLP), ANFIS, geostatistical models | NA |
| Feedforward | [122] | Tai Po River (China) | DO; pH, EC, WT, $NH_3$-N, TP, $NO_2$, $NO_3$ | NA | CL | Category 2 | 252 samples (21 years) | No details | Train:85%, test: 15% | ANN, ANFIS, MLR | NA |
| Feedforward | [137] | Ireland Rivers (Ireland) | DO, BOD, Alk, TH;; WT, pH, EC | NA | DOP (dissolved oxygen percentage), CL;; | Category 2 | 3001 samples (No details) | No details | No details | ANN | NA |
| Feedforward | [42] | Twostatistical databases (European countries) | BOD; DO | NA | other 20 variables | Category 2 | 159 samples (9 years) | No details | Train:88%, test: 12% | GRNN, MLR | NA |
| Feedforward | [81] | Maroon River (Iran) | WT, Tur, pH, EC, TDS, TH; | NA | $HCO_3$, $SO_4$, CL, Na, K, Mg, Ca | Category 2 | No details (20 years) | monthly | Train:60%, validate: 15%, test: 35% | ANN(MLP), RBFNN | NA |
| Feedforward | [36] | River Zayanderud (Iran) | TSS; pH, TH | NA | Na, Mg, $CO_3^{2-}$, $HCO_3$, CL, Ca | Category 2 | 1320 samples (10 years) | monthly | No details | RBFNN, TDNN | NA |
| Feedforward | [9] | Ardabil plain (Iran) | EC, TDS; | RF | RO, WL | Category 2 | No details (17 years) | 6 months | Train:71%, test: 29% | ANN, MLR | 1 |
| Feedforward | [25] | Danube River (Serbia) | BOD; WT, DO, pH, $NH_3$-N, COD, EC, $NO_3$, TH, TP | NA | other 8 variables | Category 2 | more than 32,000 samples (years) | No details | Train:72%, validate: 18%, test: 10% | GRNN | NA |

Table A1. *Cont*.

| Categories | Authors (Year) | Locations | Water Quality Variables | Meteorological Factors | Other Factors | Output Strategy | Dataset | Time Step | Data Dividing | Methods | Prediction Lengths |
|---|---|---|---|---|---|---|---|---|---|---|---|
| Feedforward | [82] | Hydrometric stations (USA) | SS;; | NA | Q | Category 0 and Category 3 | No details (8 years) | daily | Train and test:80%, validate:20% | ANN(MLP), SVR, MLR | 1 |
| Feedforward | [138] | Surma River (Angladesh) | BOD, COD;; | NA | NA | Category 0 and Category 3 | No details (3 years) | No details | Train:70%, validate: 15%, test: 15% | RBFNN, MLP | NA |
| Feedforward | [85] | Groundwater (Palestine) | S; EC, TDS, NO$_3$ | NA | Mg, Ca, Na | Category 2 | No details (11 years) | No details | Train: more than 50%, test: less than 50% | ANN(MLP), SVM | NA |
| Feedforward | [24] | River Danube (Hungary) | DO; pH, WT, EC | NA | RO | Category 2 | More than 151 samples (6 years) | monthly | No details | GRNN, ANN(MLP), RBFNN, MLR | NA |
| Feedforward | [60] | Langat River and Klang River (Malaysia) | DO, BOD, COD, SS, pH, NH$_3$-N; | NA | NA | Category 2 | No details (10 years) | monthly | Train:80%, validate: 20% | RBFNN | NA |
| Feedforward | [47] | Eight United States Geological Survey stations (USA) | DO; WT, EC, Tur, pH | NA | YMDH | Category 2 | 35,064 samples (4 years) | hourly | Train:70%, test: 30% | ELM, ANN(MLP) | 1, 12, 24, 48, 72, 168 |
| Feedforward | [162] | Rivers (China) | DO; WT, pH, BOD, NH$_3$-N, TN, TP | NA | other variables | Category 2 | 969 samples | No details | Train and validate: 80%, test: 20% | BPNN, SVM, MLR | NA |
| Feedforward | [86] | Syrenie Stawy Ponds (Poland) | DO, BOD, COD, TN, TP, TA | NA | CL; other ions | Category 2 | No details (19 months) | monthly | Train:60%, validate: 20%, test: 20% | ANN(MLP) | NA |
| Feedforward | [83] | Delaware River (USA) | DO; pH, EC, WT | NA | Q | Category 1 and Category 2 | 2063 samples (6 years) | daily | Train:75%, test: 25% | ANN(MLP), RBFNN, SVM | NA |
| Feedforward | [84] | Zayandeh-rood River (Iran) | NO$_3$; EC, pH, TH | NA | Na, K, Ca, Mg, SO$_4$, CL, bicarbonate | Category 2 | No details | No details | Train:50%, validate: 30%, test: 20% | ANN(MLP) | NA |

Table A1. *Cont.*

| Categories | Authors (Year) | Locations | Water Quality Variables | Meteorological Factors | Other Factors | Output Strategy | Dataset | Time Step | Data Dividing | Methods | Prediction Lengths |
|---|---|---|---|---|---|---|---|---|---|---|---|
| Feedforward | [59] | Saint John River (Canada) | TSS, COD, BOD, DO, Tur; | NA | NA | Category 2 | 39 samples (3 days) | No details | Train:60%, validate: 20%, test: 20% | BPNN, SVM | NA |
| Feedforward | [164] | Karkheh River (Iran) | BOD; TDS, EC | NA | CL, Na, $SO_4$, Mg, SAR, Ca | Category 2 | 13,800 samples (5 years) | No details | No details | ANN | NA |
| Feedforward | [159] | Xuxi River (China) | COD; WT, DO, TN, TP, $NH_3$-N, SD, SS | NA | NA | Category 2 | 110 samples (8 hours) | No details | No details | MLP | NA |
| Feedforward | [102] | Danube River (Serbia) | DO; pH, WT, EC, BOD, COD, SS, P, $NO_3$, TA, TH | NA | five metal ions | Category 2 | No details (6 years; 7 years) | monthly or fortnightly | Train:72%, validate: 18%, test: 10% | BPNN | NA |
| Feedforward | [131] | Sufi Chai river (Iran) | TDS; | NA | Q, Other 4 variables | Category 2 | 144 samples (12 years) | monthly | Train:66%, validate: 17%, test: 17% | ANN(MLP) | NA |
| Feedforward | [127] | River Tisza (Hungary) | DO; WT, EC, pH | NA | RO | Category 2 | More than 1300 samples (6 years) | No details | Train:67%, test: 33% | RBFNN, GRNN, MLR | 12 |
| Feedforward | [171] | Karoon River (Iran) | TH; EC, TDS, pH | NA | SAR; $HCO_3$, CL, $SO_4$, Ca, Mg, Na, K, TAC | Category 2 | No details (49 years) | No details | No details | ANN(MLP), RBFNN | NA |
| Feedforward | [32] | Yamuna River (India) | DO;; BOD, COD, pH, WT, $NH_3$-N | NA | Q | Category 3 | No details (4 years) | monthly | Train:75%, test: 25% | BPNN, SVM, ANFIS, ARIMA | NA |
| Feedforward | [88] | Lakes (USA) | Chl-a; TP, TN, Tur | NA | SD | Category 2 | 1087 samples (6 years) | No details | Train:75%, test: 25% | MLP, ANFIS | NA |
| Feedforward | [139] | Karoun River (Iran) | BOD, COD; EC, Tur, pH | NA | six mental ions | Category 2 | 200 samples (16 years) | No details | No details | ANN, ANFIS, Least Squares SVM(LSSVM) | NA |

**Table A1.** *Cont.*

| Categories | Authors (Year) | Locations | Water Quality Variables | Meteorological Factors | Other Factors | Output Strategy | Dataset | Time Step | Data Dividing | Methods | Prediction Lengths |
|---|---|---|---|---|---|---|---|---|---|---|---|
| Feedforward | [133] | Lakes (USA) | TN, TP; pH, EC, Tur | NA | NA | Category 2 | 1217 samples | No details | Train:55%, validate: 22%, test: 23% | ANN, LR | NA |
| Feedforward | [48] | Three rivers (USA) | WT; | AT | Q, DOY | Category 2 | No details (8 years) | No details | No details | ELM, ANN(MLP), MLR | NA |
| Feedforward | [63] | St. Johns River (USA) | DO; $NH_3$-N, TDS, pH, WT | NA | CL | Category 2 | 232 samples (12 years) | half a month | Train:75%, test: 25% | CCNN, DWT, VMD-MLP, MLP | NA |
| Recurrent | [111] | Talkheh Rud River (Iran) | TDS; | NA | Q | Category 1 | No details (13 years) | No details | Train:69%, validate & test: 31% | Elman, ANN(MLP) | 1 |
| Recurrent | [3] | Hyriopsis Cumingii ponds (China) | DO;; pH, WT | SR, WS, AT | NA | Category 3 | 816 samples (34 days) | No details | Train and validate:80%, test: 20% | Elman | NA |
| Recurrent | [41] | Danube River (Serbia) | DO; WT, pH, EC | NA | Q | Category 2 | 61 samples | monthly or semi-monthly | Train: 85%, test: 15% | Elman, GRNN, BPNN, MLR | NA |
| Recurrent | [167] | Chou-Shui River (China) | pH, Alk | NA | As;; Ca | Category 3 | No details (8 years) | No details | No details | Systematical dynamic-neural modeling (SDM), BPNN, NARX | NA |
| Recurrent | [55] | Yenicaga Lake (Turkey) | DO; WT, EC, pH | NA | WL, DOY, hour | Category 2 | 13,744 samples (573 days) | 15 minutes | Train:60%, validate: 15%, test: 25% | TLRN, RNN, TDNN | NA |
| Recurrent | [12] | Dahan River (China) | TP;; EC, SS, pH, DO, BOD, COD, WT, $NH_3$-N | NA | Coli | Category 3 | 280 samples (11 years) | monthly | Train:75%, test: 25% | NARX, BPNN, MLR | 1 |

**Table A1.** *Cont.*

| Categories | Authors (Year) | Locations | Water Quality Variables | Meteorological Factors | Other Factors | Output Strategy | Dataset | Time Step | Data Dividing | Methods | Prediction Lengths |
|---|---|---|---|---|---|---|---|---|---|---|---|
| Recurrent | [6] | Taihu Lake (China) | DO, TP;; | NA | NA | Category 0 | 657 samples (7 years) | monthly | Train:90%, test: 10% | LSTM, BPNN, OS-ELM | NA |
| Recurrent | [38] | WWTP(China) | BOD, TP;; COD, TSS, pH, DO, WT | NA | ORP | Category 2 and Category 3 | 5000 samples | No details | Train:45%, validate: 15%, test: 40% | RESN | NA |
| Recurrent | [66] | Mariculturebase (China) | WT, pH; EC, S, Chl-a, Tur, DO | NA | NA | Category 2 | 710 samples (21 days) | 5 minutes | Train:86%, test: 14% | LSTM, RNN | >32 |
| Recurrent | [67] | Marine aquaculture base (China) | pH, WT;; | NA | NA | Category 0 | 710 samples | No details | Train:86%, test: 14% | SRU | NA |
| Recurrent | [53] | Geum River basin (Korea) | BOD, COD, SS; | AT, WS | WL, Q | Category 2 | No details (10 years) | daily | Train:70%, test: 30% | RNN, LSTM | 1 |
| Recurrent | [165] | Lakes (USA) | WT;; | NA | NA | Category 0 | 1520 samples | No details | Train:65%, test: 35% | LSTM | NA |
| Recurrent | [153] | Reservoir (China) | Chl-a;; WT, pH, EC, DO, Tur | NA | ORP | Category 0 and Category 2 | 1440 samples (5 days) | 5 minutes | No details | TL-FNN, RNN, LSTM | NA |
| Recurrent | [134] | Two gauged stations (USA) | SS;; | NA | Q | Category 1 | 10,060 samples (30 years) | daily | Train: 70–90%, test: 30–10% | WANN | NA |
| Recurrent | [135] | Agricultural catchment (France) | NO$_3$, SS; | RF | Q | Category 1 and Category 2 | 26,355 samples (1 year) | daily | Train: 66.67%, test: 33.33% | SOM-MLP, MLP | NA |
| Recurrent | [140] | Four streams (USA) | WT; | SR, AT | NA | Category 2 | No details (4 years) | 10 minutes | Train:50%, validate: 25%, test: 25% | u GA-ANN, BPNN, RBFNN | NA |
| Hybrid | [141] | Chaohu Lake (China) | TP, TN, Chl-a; | NA | Bands | Category 2 | 18,368 (TN),1050(TP) samples (more than 3 years) | No details | Train:86%, test: 14% | GA-BP, BPNN, RBFNN | NA |
| Hybrid | [142] | Two stations (USA) | SS;; | NA | Q | Category 1 and Category 3 | 730 samples (2 years) | daily | Train:50%, test: 50% | ANN-differential evolution | NA |

**Table A1.** *Cont.*

| Categories | Authors (Year) | Locations | Water Quality Variables | Meteorological Factors | Other Factors | Output Strategy | Dataset | Time Step | Data Dividing | Methods | Prediction Lengths |
|---|---|---|---|---|---|---|---|---|---|---|---|
| Hybrid | [71] | B¨uy¨uk Menderes river (Turkey) | WT, DO, B;; | NA | NA | Category 0 | 108 samples (9 years) | monthly | Train:67%, test: 33% | ARIMA-ANN, ANN, ARIMA | NA |
| Hybrid | [143] | Karkheh reservoir (Iran) | water quality variables | NA | NA | Category 2 | No details (6 months) | No details | No details | PSO-ANN | NA |
| Hybrid | [1] | WWTP(China) | DO; COD, BOD, SS | NA | other two variables | Category 3 | No details | daily | No details | SOM-RBFNN, ANN(MLP) | NA |
| Hybrid | [144] | Bangkok canals (Thailand) | DO;; WT, pH, BOD, COD, SS, $NH_3$-N, TP, $NO_2$, $NO_3$, | NA | total coliform, hydrogen sulfide | Category 3 | 13,846 samples (5 years) | monthly | Train: 70%, test: 30% | FCM-MLP, MLP | 1 |
| Hybrid | [56] | Lake Baiyangdian (China) | Chl-a; WT, pH, DO, SD, TP, TN, $NH_3$–N, BOD, COD | Precip, Evap | WL, LV, Sth | Category 2 | No details (10 years) | monthly | No details | WANN, ANN, ARIMA | NA |
| Hybrid | [64] | Songhua River (China) | DO, $NH_3$-N;; | NA | NA | Category 0 | No details (7 years) | monthly | Train:71%, test: 29% | BWNN, ANN, WANN, ARIMA | 1 |
| Hybrid | [136] | Gazacoastal aquifer (Palestine) | $NO_3$; EC, TDS, $NO_3$, | | CL, $SO_4$, Ca, Mg, Na | Category 2 | No details (10 year) | No details | No details | K-means-ANN | NA |
| Hybrid | [43] | WWTP (Turkey) | COD; SS, pH, WT | NA | Q | Category 2 | 265 samples (3 years) | daily | Train:50%, validate:25%, test: 25% | k-means-MLP, Arima-RBF, ANN(MLP), MLR, RBFNN, GRNN, ANFIS | NA |
| Hybrid | [70] | Yangtze River (China) | DO, $NH_3$-N;; | NA | NA | Category 0 | 480 samples (9 years) | weekly | Train:67%, validate & test: 33% | ARIMA-RBFNN | 1 |
| Hybrid | [120] | Taihu Lake (China) | DO, EC, pH, $NH_3$-N, TN, COD, TP, BOD, COD; | NA | VP, petroleum, other 11 variables | Category 2 | 2680 samples | No details | Train:75%, test: 25% | PCA-GA-BPNN | NA |

**Table A1.** *Cont.*

| Categories | Authors (Year) | Locations | Water Quality Variables | Meteorological Factors | Other Factors | Output Strategy | Dataset | Time Step | Data Dividing | Methods | Prediction Lengths |
|---|---|---|---|---|---|---|---|---|---|---|---|
| Hybrid | [62] | Gauging station (Iran) | DO, WT, S;; Tur, Chl-a | NA | NA | Category 0 and Category 2 and Category 3 | 650, 540 samples | daily, hourly | Train:70%, validate: 15%, test: 15% | WANN, ANN | 1, 2, 3 |
| Hybrid | [172] | Two gauging stations (USA) | SS;; | NA | Q | Category 0 and Category 3 | 1974 samples (8 years) | daily | Train:75%, test: 25% | WANN | NA |
| Hybrid | [173] | River Yamuna (India) | COD;; | NA | NA | Category 0 | 120 samples (10 years) | monthly | Train:92.5%, test: 7.5% | ANN, ANFIS, WANFIS | 9 |
| Hybrid | [100] | Two catchments (Poland) | WT; | AT | Q, declination of the Sun | Category 2 | No details (10 years) | daily | No details | MLP, ANFIS, WNN, Product-Unit ANNs (PUNN), ensemble aggregation approach | 1, 3, 5 |
| Hybrid | [7] | South San Francisco bay (USA) | Chl-a;; | NA | NA | Category 0 | No details (20 years) | monthly | Train:60%, validate: 20%, test: 20% | WANN, MLR, GA-SVR | 1 |
| Hybrid | [72] | Asi River (Turkey) | EC;; | NA | Q | Category 0 and Category 3 | 274 samples (23 years) | No details | Train:75%, test: 25% | WANN, ANN | NA |
| Hybrid | [146] | Klamath River (USA) | DO;; pH, WT, EC, SD | NA | NA | Category 0 and Category 2 | No details | monthly | Train:80%, validate: 10%, test: 10% | WANN, ANN, MLR | NA |
| Hybrid | [147] | Prawn culture ponds (China) | WT; | NA | NA | Category 0 | 1152 samples (8 days) | 10 minutes | Train:87.5%, test: 12.5% | EMD-BPNN, BPNN | 1 |
| Hybrid | [44] | WWTP(China) | BOD; COD, SS, DO, pH | NA | NA | Category 2 | 598 samples (19 months) | daily | No details | Chaos Theory-PCA-ANN | NA |

**Table A1.** *Cont.*

| Categories | Authors (Year) | Locations | Water Quality Variables | Meteorological Factors | Other Factors | Output Strategy | Dataset | Time Step | Data Dividing | Methods | Prediction Lengths |
|---|---|---|---|---|---|---|---|---|---|---|---|
| Hybrid | [174] | Charlotte harbor marine waters | TN; | NA | NA | Category 0 | No details (13 years) | monthly | Train:70%, validate: 15%, test: 15% | WANN, wavelet-gene expression programing (WGEP), TDNN, GEP, MLR | 1 |
| Hybrid | [73] | Groundwater (Iran) | EC, Tur, pH, $NO_2$, $NO_3$ | NA | Cu | Category 2 | No details (8 years) | No details | Train:80%, test: 20% | PCA-ANN | NA |
| Hybrid | [17] | Downstream (China) | WT, DO, pH, EC, TN, TP, Tur, Chl-a; | NA | NA | Category 0 | No details (13 months) | daily | Train:80%, validate: 10%, test: 10% | Ensemble-ANN | 1 |
| Hybrid | [104] | Karaj River (Iran) | $NO_3$; | NA | CL; Q | Category 0 and Category 1 and Category 3 | No details | monthly | Train:80%, validate: 10%, test: 10% | WANN, ANN, MLR | NA |
| Hybrid | [148] | Crab ponds (China) | DO;; WT | SR, WS, AT, AH | NA | Category 3 | 700 samples (22 days) | 20 minutes | Train:71%, test: 29% | RBFNN-IPSO-LSSVM, BPNN | 3 |
| Hybrid | [149] | Guanting reservoirs (China) | DO, COD, $NH_3$-N;; | NA | NA | Category 0 | No details (18 weeks) | weekly | No details | Kalman-BPNN | 2 |
| Hybrid | [101] | Toutle River (USA) | SS;; | NA | Q | Category 0 and Category 3 | 2000 samples (8 years) | daily | No details | A least-square ensemble models-WANN | NA |
| Hybrid | [69] | WWTP (China) | DO; pH | NA | NA | Category 2 | 50 samples | No details | Train:70%, test: 30% | FNN-WNN | NA |
| Hybrid | [52] | Clackamas River (USA) | DO;; WT | NA | Q | Category 3 | 1623 samples (6 years) | daily | Train:78%, test: 22% | WANN, WMLR, ANN(MLP), MLR | 1, 31 |

**Table A1.** *Cont.*

| Categories | Authors (Year) | Locations | Water Quality Variables | Meteorological Factors | Other Factors | Output Strategy | Dataset | Time Step | Data Dividing | Methods | Prediction Lengths |
|---|---|---|---|---|---|---|---|---|---|---|---|
| Hybrid | [123] | Representative lakes (China) | Chl-a; WT, pH;; $NH_3$-N, TN, TP, DO, BOD | NA | other 17 variables | Category 3 | No details (3 years) | No details | Train:80%, test: 20% | GA-BP | NA |
| Hybrid | [16] | Miyun reservoir (China) | DO, COD, $NH_3$-N; | NA | NA | Category 0 | 5000 samples (2 years) | weekly | Train:98%, test: 2% | PSO-WNN, WNN, BPNN, SVM | NA |
| Hybrid | [126] | Aji-Chay River (Iran) | EC;; | NA | NA | Category 0 | 315 samples (26 years) | monthly | Train:90%, test: 10% | WA-ELM, ANFIS | 1, 2, 3 |
| Hybrid | [4] | Yangtze River (China) | DO, $COD_{Mn}$, BOD;; | NA | NA | Category 3 | 65 samples (2 months) | daily | Train:50%, validate: 16%, test: 34% | IABC-BPNN, BPNN | NA |
| Hybrid | [33] | WWTP(China) | COD; COD, SS, pH, $NH_3$-N | NA | NA | Category 2 | 250 samples | No details | No details | WANN, ANN(MLP) | NA |
| Hybrid | [175] | The Stream Veszprémi-Séd (Hungary) | pH, EC, DO, Tur;; | NA | NA | Category 2 | No details (7 years) | yearly | No details | DE-ANN | NA |
| Hybrid | [54] | Shrimp pond (China) | DO; WT, $NH_3$-N, pH | AT, AH, AP, WS | NA | Category 2 | 2880 samples (20 days) | 10 minutes | Train:75%, test: 25% | SAE-LSTM, SAE-BPNN, LSTM, BPNN | 18, 36, 72 |
| Hybrid | [124] | Four basins (Iran) | TDS; EC | NA | Na, CL | Category 2 | No details (20 years) | No details | Train:80%, test: 20% | WANN, GEP, WANFIS | NA |
| Hybrid | [125] | Blue River (USA) | pH, DO, Tur; WT | NA | Q | Category 0 and Category 3 | No details (4 years) | daily | Train:80%, test: 20% | WANN, WGEP | 1 |
| Hybrid | [157] | Chattahoochee River (USA) | pH;; | NA | Q | Category 3 | 730 samples (2 years) | daily | Train:75%, test: 20% | WANN, ANN, WMLR, MLR | 1, 2, 3 |
| Hybrid | [176] | Morava River Basin (Serbia) | WT, EC; SS, DO | NA | other ions | Category 2 | No details (10 years) | 15 days | No details | PCA-ANN | NA |
| Hybrid | [151] | Tai Lake, Victoria Bay (China) | DO;; WT, pH, $NO_2$, TP | Precip | NA | Category 3 | No details (7 years) | No details | Train:80%, test: 20% | IGRA-LSTM, BPNN, ARIMA | NA |

**Table A1.** *Cont.*

| Categories | Authors (Year) | Locations | Water Quality Variables | Meteorological Factors | Other Factors | Output Strategy | Dataset | Time Step | Data Dividing | Methods | Prediction Lengths |
|---|---|---|---|---|---|---|---|---|---|---|---|
| Hybrid | [46] | WWTP (Saudi Arabia) | C, DO, SS, pH | NA | CL;; | Category 3 | 774 samples | No details | No details | PCA-ELM | NA |
| Hybrid | [5] | Prespa Lake (Greece) | DO, Chl-a;; | NA | NA | Category 0 | 363 samples (11 months) | daily | Train:70%, validate: 15%, test: 15% | CEEMDAN-VMD-ELM) | NA |
| Hybrid | [87] | The Warta River (Poland) | WT;; | AT | NA | Category 3 | No details (22 to 27 years) | daily | Train:4/9, validate: 2/9, test: 1/3 | WANN(MLP), MLP | 1 |
| Hybrid | [152] | Ashi River (China) | DO, NH$_3$-N, Tur;; | NA | NA | Category 0 | 846 samples (4 hours) | more than 4 months | Train:70%, test: 30% | IGA-BPNN | 1 |
| Hybrid | [15] | Qiantang River (China) | pH, TP, DO;; | NA | NA | Category 0 | 1448 samples | No details | Train:70%, test: 30% | DS-RNN, RNN, BPNN, SVR | NA |
| Hybrid | [132] | The Johor river (Malaysia) | NH$_3$-N, SS, pH; Tur, WT, | NA | COD$_{Mn}$, Mg, Na | Category 2 | No details (1 year) | No details | No details | WANFIS, MLP, RBFNN, ANFIS | NA |
| Hybrid | [103] | Hilo Bay (the Pacific Ocean) | Chl-a, S;; | NA | NA | Category 0 | No details (5 years) | daily | No details | Bates–Granger (BG)-least square based ensemble (LSE)-WANN | 1, 3, 5 |
| Hybrid | [154] | WWTP (China) | COD, TP, pH, TN; DO, NH$_3$-N, BOD, TH | NA | CL, oil-related quality indicators | Category 2 | 23,268 samples (4 years) | hourly | Train:80%, test: 20% | PSO-LSTM | 1 |
| Hybrid | [68] | Beihai Lake (China) | pH, Chl-a, DO, BOD, EC; | NA | HA;; | Category 3 | No details (5 days) | 30 minutes | Train:70%, test: 30% | PSO-GA-BPNN | 12 |
| Hybrid | [26] | River (China) | COD;; | NA | NA | Category 0 | 460 samples (14 months) | 12 hours | Train:95%, test: 5% | LSTM-RNN | 1 |

**Table A1.** *Cont.*

| Categories | Authors (Year) | Locations | Water Quality Variables | Meteorological Factors | Other Factors | Output Strategy | Dataset | Time Step | Data Dividing | Methods | Prediction Lengths |
|---|---|---|---|---|---|---|---|---|---|---|---|
| Hybrid | [45] | Zhejiang Institute of Freshwater Fisheries (China) | DO; WT | AT, AH, WS, WD, SR, AP | SM, ST | Category 4 | 5006 samples (1 year) | 10 minutes | Train:80%, test: 20% | attention-RNN | 6, 12, 48, 144, 288 |
| Hybrid | [39] | Taihu Lake (China) | pH; DO, COD, NH$_3$-N | NA | NA | Category 2 | 28 samples (6 months) | Weekly | Train:75%, test: 25% | grey theory-GRNN, BPNN, RBFNN | 1 |
| Emerging | [58] | Wastewater factory (China) | TP; WT, TSS, pH, NH$_3$-N, NO$_3$, DO | NA | other 3 variables | Category 2 | 1000 samples (4 months) | No details | Train:80%, test: 20% | SODBN | NA |
| Emerging | [57] | Recirculating Aquaculture Systems (China) | DO;; EC, pH, WT | NA | NA | Category 3 | 4500 samples (13 months) | 10 minutes | Train:67%, validate: 11%, test: 22% | CNN, BPNN | 18 |

The contents before the ";" symbol were the output variables; The contents before the ";;" symbol were output and predictors; NA represents blank content.

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
