# Peer review of "A Review of the Artificial Neural Network Models for Water Quality Prediction"

_applsci, doi:10.3390/app10175776_

Round 1
Reviewer 1 Report
Dear authors
Thank you for a great attempt to make a review of all ANN methods in the field of review. It is an ambitious attempt.
As a reader, I always expect more analysis in the review papers compared to the specific topic research. A review, in my opinion, should always provide discussion or a strategy to a future development, shortage of any point and authors’ analysis, including development, trials and future strategies. It is untrue to say that you did not give any of the listed; however, all points made in the review are just listing of all conclusions from other papers without deep discussion or analysis. Without it, this paper feels shallow although it is quite long.
Some of your statements have no support in the text, for example: “It can be concluded without a doubt that the future is bright for the ANN model’s applications in water quality”. You base your thinking on the number of publication or results ANN produces in water quality. What is a merit of such statement? In your analysis, there are 107 papers without description of the prediction length, and 30 with just a day prediction. I am not sure, if you can conclude “future is bright” from this.
Some sentences just needs to be re-written. Example “majority of the ANNs methods are used to predict the DO, BOD, COD, chemical variables, which verify the importance of them as a primary indicator of the ecosystem” – if majority uses these variables to predict it doesn’t mean they are verified important.
Please, differentiate river, water reservoir and artificial facilities in water quality prediction, which ANN type could provide better analysis, which methods shows significant results, how you address different parameters. This could be easily picked from your tables, which I find useful, but they still need lots of work to improve. I don’t see how by reading this review, reader should decide what questions are answered, which are still in progress, and to which you should pay attention.
ANN structures, features and technical differences are well described; however, very few details of the impacts to the water quality parameters are given. Please, write more about applications of the ANN, not ANN itself.
Although, I am not a native speaker this paper is really hard to read, please consider revising and proofreading it.
Some small comments
How you chose typical reference in your tables? Why not list several ones?
“A given study” is usually used to talk about manuscript itself, not a study reference. Example: We specifically address groundwater and river systems in the given study.
What worthy of study, learning does mean? You are supposed to give characteristics based on the review, this is just a suggestion but without your reasoning.
101-102 The popularity of ANNs above is also an agreement with the observation by other researches [10,14]. Probably – in agreement.
112-113 the central nervous systems of animals of biological neural networks – please modify.
132 Figure 2 is missing RNN – Should probably give a full name here or separate 4 architectures in table
134 Table 2. I would suggest improving table 2, head of the table and content is not sometime, what it is supposed to be. Example: in structure of LSTM you state – an improvement of the RNN, this is not structure, it is more description
What is DBN?
150 mistype pattern
164 strange sentence construction
Please, consider improving writing of 2.2, 2.3 and partially 2.4 paragraphs
236 mistype are discussed
245 what do you mean by commonly?
246 In above case – which case?
247 Which studies – reference – combine with 248
250-251 – please, re-write this
253 What do you mean by primary indicator?
255-257 please, re-write this
In Figure 9, you show that DO variable prediction is used in 58 paper, however, in the table 3 there is only one reference, same for other parameters e.g. BOD, COD and etc.. How you chose this reference, and why you don’t list all?
264 what other water quality means?
274-275 please, re-write this
275-276 please, re-write this
281-282 please, re-write this
306 – which ones?
Table 3, how you select reference field here? Is it only one research? The most significant of them? What is criteria?
321 please, re-write this
324 please, re-write this
330-332 It is not pleasing to see – is strange phrase choice, please reconsider
335-336 You can list 3 model types here, as Figure 8 too far from the text? Especially so as you mention this model below in next sentences.
338 external attributes 339 external variables – please choose one
Figure 10. What black dots represents here?
349 respectively, not needed
359 – [127] studies the spatial relationship between other variables. And, what are your thoughts?
362 18 papers using multiple output strategies. Why this is important?
380 declines
382 Figure 11 is not informative, what is forecast length for 32? Why don’t you use just days of forecast instead?
436 Table 8. What does Missing data imputation – No details or only mentioned – Not recommend – means? What information is given here?
442-444 Strange usage of phrase “It’s a poor situation”, please rewrite
462 are presented
467 So “ANN model is superior because of the satisfactory performance” but why it provides satisfactory performance is more important.
470 ”may” should be “maybe”
472 – repetition
523 Karaboga please give reference to his work(s)
529-531 “so the total number of times is more than 151” – could not understand meaning of this. Number of good performance, number of comparisons, number of articles….
541 should be - may be
570 A “is” a constant
575-577 please, re-write this
582-588 please, re-write this
605-606 please, re-write this
611 is that – mistype
616 Adam-Optimize - I believe it is Adam optimization
623 river – two times
626 importance of them as a primary indicator of the ecosystem – where you come up from for this? It doesn’t mean that. Although, this parameters are very important but sentence constriction and logic is wrong.
627-628 Probably, you mean - the main variables measured by sensors are
637-639 – please, re-write this sentence
660-661– are
665-666 In reviewed papers, not this paper
666-667 This statement is not supported by any outcome. They can further study, but what is merit, which one suits better, what are results of your analysis?
Author Response
Dear reviewer,
Thank you for taking the time to read my review and give me lots of valuable advice.
The attachment is my specific response to your comments.
Please don't hesitate to let us know if you have any questions about this.
Thank you in advance for your time.
Sincerely,
August 1, 2020
Authors: Yingyi Chen, Lihua Song, Yeqi Liu, Ling Yang, Daoliang Li.
Corresponding author: Yingyi Chen (Prof. & Dr.)
China Agricultural University, Beijing, 100083. P. R. China
Tel: +86-10-6273 8489
Fax: +86-10-6273 7741
Email: chenyingyi@cau.edu.cn

Reviewer 2 Report
The authors made a considerable review work but the paper needs some improvement in order to be published in an international journal.
At first the paper is definitively too long: the authors should focus more on aspects related to the application of methods to the water quality sector, while they use many space to the specification of the different models. A synthesis must be done, focusing more on the Discussion section, which should be the hearth of the work itself.
Then, the introduction section lacks in being a proper introduction to the work: authors should focus on water quality assessment-modeling-prediction using proper references. In some cases some assertions have been done without proper citations or explanation: I have marked in the annotated version of the paper.
Further, the whole structure of the paper needs to be revised, including a Methods section and properly discerning between the review of models and the comments/evaluation. Such a kind of review work needs to be perfectly structured, reaching a clearness that doesn't let the reader looking annoying but finding directly features of the revised papers according to the water quality issue, and related comments.
English language must be improved throughout the whole paper, correcting errors and improving style and readability.
Finally, many problems needs to be solved in formatting and in properly sequencing of figures and tables: figures and tables must be presented when they are recalled in the text and then correctly numbered.
In the attached version authors may find points to be clarified and/or corrected.

Author Response

(The authors gave the same response as above.)

Reviewer 3 Report
The article presents a review of the Artificial Neural Network (ANN) models for water quality prediction. The paper focus on an interesting issue and research results can be valuable. The paper is suitable topic for the readership of Applied Sciences MDPI Journal. Therefore, I recommend a revision of this manuscript considering my comments below.
My comments are as follows:
- Lines 46-48: Confusing sentence. Please rephrase it;
- Introduction: As an Introduction for a Review paper, I would advise the authors to use more references;
- Figure 3 has low resolution;
- Line 144: Correct “MPL”;
- Lines 335-342: These five categories should be better explained/detailed, in addition to the information given in table 6;
- Lines 362-363: Complete or correct the sentence “18 papers using multiple output strategies.”
- Lines 394-397: Confusing sentence. Please revise it;
- Lines 399-401: These sentences are also confusing. Please revise them;
- Line 405: Explain “overfitting”;
- Lines 413-414: Confusing sentence. Please revise it;
- Table 8 can be more discussed;
- Lines 456-459: Confusing sentence.
- Lines 466-467: How can the authors affirm that? It should be at least referenced.
- Table 9: Although this term is broadly known, “sqrt” has not been defined;
- Line 623: The word “river” is repeated;
- Discussion: This section could be concluded with the contribution of this paper for the literature.
Author Response

(The authors gave the same response as above.)

Round 2
Reviewer 1 Report
Dear Authors,
The review looks much better and I really like how you reorganized it.
There is only one chapter that I would like you to improve: 4. Artificial neural networks models for water quality prediction.
It is very long and hard to follow. Currently, it is just a list of the works from reference with just one-two sentences of description, for example: "368 Yang et al [142] found the most significant parameters by using analysis of variance (ANOVA) techniques" (you don't list the parameters here, why?).
There are two options how to divide it to sub-sections. First using figure 1 (model architecture) as a structure – Feedforward, Recurrent, Hybrid, Emerging methods. This will give you possibility to show shortcomings and advantages in each methods through the links to the proper papers.
Second, to organize this chapter on the basis of the prediction of the measured values (using sensors) and prediction of the values that can’t be express measured (BOD, for example). Thus, reader will be able to define variables that can be measured and which methods provides suitable solution; variables that can’t be measured and which methods and options should be chosen.
Of course, some of the ideas are listed further in the results section, but I believe by dividing and reclassifying section 4 you will significantly improve this review.
124 - Please, add that the Table 2 lists representative reference.
Author Response
Dear reviewer,
Thank you for taking the time to read my review and give me lots of valuable advice.
The following attachments are my responses to your comments.
Please don't hesitate to let us know if you have any questions about this.
Thank you in advance for your time.
Sincerely,
August 14, 2020
Authors: Yingyi Chen, Lihua Song, Yeqi Liu, Ling Yang, Daoliang Li.
Corresponding author: Yingyi Chen (Prof. & Dr.)
China Agricultural University, Beijing, 100083. P. R. China
Tel: +86-10-6273 8489
Fax: +86-10-6273 7741
Email: chenyingyi@cau.edu.cn

Reviewer 2 Report
The authors answered to the 1st round review in a quite satisfying way but some minor points still remain and are marked in the annotated version. Some crossed words are present in the paper and their presence is quite bothering.
Formatting issue still remain at the end of the paper, both in the text and in the table. The text, as it is actually interrupted, is not clear if it is complete or not: maybe a finale remarks still lacks.
Even "Author Contributions", "Acknowledgments" and "Conflicts of Interest" have been cancelled.

Author Response

(The authors gave the same response as above.)
